# Protein Circuit Tracing via Cross-layer Transcoders

**Darin Tsui** [1]   **Kunal Talreja** [1]   **Daniel Saeedi** [1]   **Amirali Aghazadeh** [1]

## Abstract

Protein language models (pLMs) have emerged as powerful predictors of protein structure and function. However, the computational circuits underlying their predictions remain poorly understood. Recent mechanistic interpretability methods decompose pLM representations into interpretable features, but they treat each layer independently and thus fail to capture cross-layer computation, limiting their ability to approximate the full model. We introduce ProtoMech, a framework for discovering computational circuits in pLMs using *cross-layer transcoders* that learn sparse latent representations jointly across layers to capture the model's full computational circuitry. Applied to the pLM ESM2, ProtoMech recovers 82–89% of the original performance on protein family classification and function prediction tasks. ProtoMech then identifies compressed circuits that use $<1\%$ of the latent space while retaining up to 79% of model accuracy, revealing correspondence with structural and functional motifs, including binding, signaling, and stability. Steering along these circuits enables high-fitness protein design, surpassing baseline methods in more than 70% of cases. These results establish ProtoMech as a principled framework for protein circuit tracing.

## 1. Introduction

Protein language models (pLMs) have driven rapid progress in the biosciences by learning rich statistical representations from large protein sequence databases. They now achieve strong performance across a broad range of downstream tasks, including 3D structure and function prediction (Lin et al., 2023; Hayes et al., 2025; Bhatnagar et al., 2025). These results suggest that pLMs may capture latent structural and functional motifs governing protein sequences (Rives et al., 2021; Tsui et al., 2025a; Tsui & Aghazadeh, 2024). However, the internal computational pathways, or *circuits*, responsible for these predictions remain opaque, poorly understood, and difficult to extract.

Recent progress in mechanistic interpretability, particularly through sparse autoencoders (SAEs) (Templeton et al., 2024; Gao et al., 2025), has enabled the decomposition of pLM hidden states into interpretable features (Adams et al., 2025; Simon & Zou, 2025; Walton et al., 2025a; Gujral et al., 2025; Nainani et al., 2025; Parsan et al., 2025). Steering these features has been shown to generate protein sequences with specific functional attributes (Tsui et al., 2025b; Corominas et al., 2025; Garcia & Ansuini, 2025). However, SAEs provide only a representational factorization and do not capture the layer-to-layer transformations that capture the model's computation. Consequently, they cannot recover the full circuitry responsible for pLM predictions. Recovering such circuits requires a *replacement model* that faithfully emulates the internal computation of the original network.

In the natural language processing literature, recent efforts have aimed to construct replacement models using *transcoders* (Dunefsky et al., 2024). In contrast to SAEs, which factorize representations, transcoders approximate the functional mapping of individual transformer MLP layers by passing activations through a sparse latent bottleneck. Composing these approximations across layers yields per-layer transcoders (PLTs), which seek to reconstruct model computation from locally sparse surrogates. Yet this construction remains inherently local: approximating each layer in isolation neglects the accumulation of context and computation across depth, leading to degraded representations and unreliable circuit recovery (Ameisen et al., 2025).

In this work, we bridge the gap between interpretable feature recovery and mechanistic circuit discovery in pLMs. We introduce ProtoMech, a framework for uncovering computational circuits in pLMs using *cross-layer transcoders* (CLTs) (Ameisen et al., 2025). Similar to standard transcoders, CLTs learn an input–output mapping for each transformer MLP layer. However, rather than relying on isolated layerwise approximations, CLTs compute each layer's output as a function of the sparse latent variables from all preceding layers. By explicitly modeling these cross-layer

[1]School of Electrical and Computer Engineering, Georgia Institute of Technology, Atltanta, GA. Correspondence to: Amirali Aghazadeh <amiralia@gatech.edu>.

*Proceedings of the $43^{rd}$ International Conference on Machine Learning*, Seoul, South Korea. PMLR 306, 2026. Copyright 2026 by the author(s).

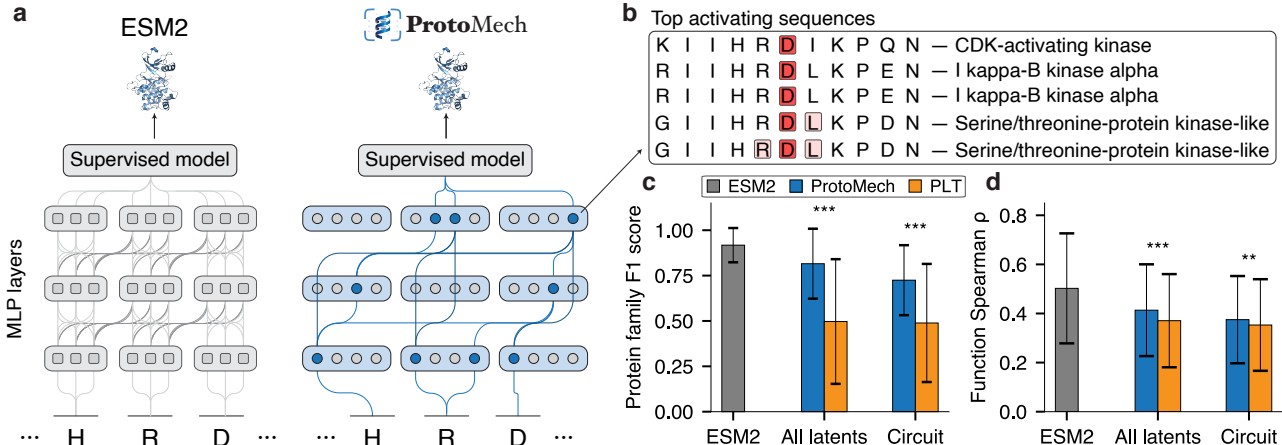

*Figure 1.* **ProtoMech serves as a replacement model for ESM2. a**, Schematic of the circuit discovery process. ProtoMech identifies a circuit of interpretable latents (blue) that traces and approximates the behavior of ESM2 on downstream tasks. **b**, Example of top activating sequences in Swiss-Prot for a specific latent (L3/1918), which detects the conserved HRD catalytic motif found in protein kinases. On **c**, protein family classification and **d**, function prediction downstream tasks, ProtoMech outperforms PLT baselines.

dependencies, ProtoMech constructs a replacement model that more faithfully reproduces the internal computation of the pLM (Fig. 1), enabling direct identification of the circuits that govern its predictions.

Our contributions are as follows:

- We develop ProtoMech, a framework for discovering and analyzing computational circuits within and across transformer layers in pLMs. Applied to ESM2 (Lin et al., 2023), ProtoMech achieves state-of-the-art recovery of the original model's performance, attaining 89% and 82% on protein family classification and function prediction tasks, respectively.

- We identify highly compact circuits that retain 79% (family) and 74% (function) of model performance while using only <1% of ProtoMech's latent space. Steering along these circuits enables the generation of a diverse set of functional sequences, including the top-fitness variants in 71% of cases.

- By analyzing circuits associated with kinase activity, NADP+ binding, and GB1, we show that ProtoMech recovers computational pathways that align with known structural and functional motifs. To facilitate broader circuit discovery, we release ProtoMech as open-source software and provide accompanying visualization tools.[1]

## 2. The ProtoMech Framework

In this section, we present the ProtoMech framework. ProtoMech is built around four interconnected components: *(i)*

[1]Our code can be found at https://github.com/amirgroup-codes/ProtoMech, and our visualizer is available at https://protmech.github.io/.

cross-layer transcoders (CLTs) for constructing a replacement model, *(ii)* a circuit discovery algorithm for identifying computational pathways, *(iii)* a steering mechanism for manipulating these circuits, and *(iv)* visualization tools for interpreting the circuits.

### 2.1. Cross-layer Transcoders (CLTs)

CLTs extend standard transcoders by approximating the input–output mapping of each MLP layer as a function of the sparse latent variables derived from all preceding layers (Fig. 2a). This formulation replaces independent layerwise surrogates with a compositional model of inter-layer computation.

Let $L$ be the number of transformer layers and $d_{\text{model}}$ the hidden dimension of the pLM. For $\ell \in 1, \ldots, L$, we denote by $\mathbf{x}^\ell \in \mathbb{R}^{d_{\text{model}}}$ the residual stream activation prior to the MLP block at layer $\ell$. Sparsity in the latent space is enforced using a TopK activation function (Makhzani & Frey, 2013). Each residual stream activation is encoded into a latent vector $\mathbf{a}^\ell \in \mathbb{R}^{d_{\text{latent}}}$ via an encoder matrix:

$$\mathbf{a}^\ell = \text{TopK}(\mathbf{W}^\ell_{\text{enc}}(\mathbf{x}^\ell - \mathbf{b}^\ell_{\text{pre}}) + \mathbf{b}^\ell_{\text{enc}}), \quad (1)$$

where $\mathbf{W}^\ell_{\text{enc}} \in \mathbb{R}^{d_{\text{latent}} \times d_{\text{model}}}$ denotes the encoder matrix, $\mathbf{b}^\ell_{\text{pre}} \in \mathbb{R}^{d_{\text{model}}}$ is the corresponding bias term at layer $\ell$, and $\mathbf{b}^\ell_{\text{enc}} \in \mathbb{R}^{d_{\text{model}}}$ is the encoder bias at layer $\ell$. To regulate the number of active latent features, the $\text{TopK}$ operator retains only the $k$ largest-magnitude latent activations and sets all others to zero. To reconstruct the output of the MLP block at layer $\ell$, denoted $\mathbf{y}^\ell \in \mathbb{R}^{d_{\text{model}}}$, CLTs employ decoder matrices that map latent representations from preceding layers to layer $\ell$ according to:

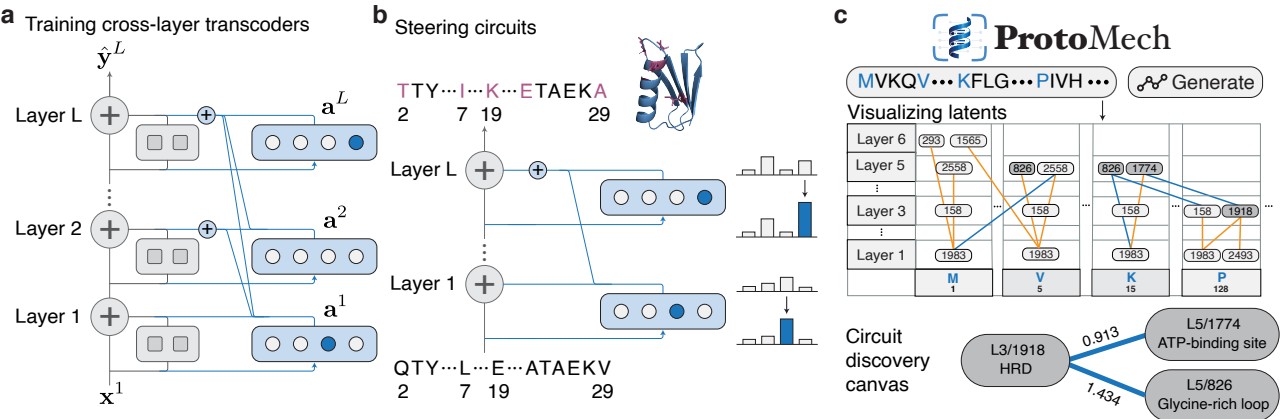

*Figure 2.* **Overview of ProtoMech. a**, Cross-layer transcoders (CLTs) form a replacement model that captures inter-layer computation by predicting each layer's output from the sparse latent features of all preceding layers. **b**, Steering along identified circuits enables the design of protein variants with enhanced functional properties. **c**, Using the ProtoMech visualizer, we expose overlapping biological motifs previously hidden in ESM2.

$$\hat{\mathbf{y}}^\ell = \sum_{\ell'=1}^{\ell} \mathbf{W}_{\text{dec}}^{\ell' \to \ell} \mathbf{a}^{\ell'} + \mathbf{b}_{\text{pre}}^\ell, \tag{2}$$

where $\hat{\mathbf{y}}^\ell \in \mathbb{R}^{d_{\text{model}}}$ denotes the reconstructed MLP output at layer $\ell$, and $\mathbf{W}_{\text{dec}}^{\ell' \to \ell}$ is a decoder matrix mapping latent features from layer $\ell'$ to layer $\ell$. By requiring each reconstruction to depend on latent representations from all preceding layers, CLTs model the accumulation of computation across depth and thereby expose the inter-layer pathways that generate the model's outputs.

We train CLTs by minimizing the mean-squared error between the original MLP output $\mathbf{y}^\ell$ and its reconstruction $\hat{\mathbf{y}}^\ell$ across all layers, $\mathcal{L}_{\text{MSE}} = \sum_{\ell=1}^{L} \|\mathbf{y}^\ell - \hat{\mathbf{y}}^\ell\|_2^2$. To mitigate the emergence of inactive ("dead") latent units, we incorporate an auxiliary loss inspired by (Gao et al., 2025). Let $\mathbf{e}^\ell = \mathbf{y}^\ell - \hat{\mathbf{y}}^\ell$ denote the reconstruction residual at layer $\ell$. The auxiliary loss is then defined as: $\mathcal{L}_{\text{aux}} = \sum_{\ell=1}^{L} \|\mathbf{e}^\ell - \hat{\mathbf{e}}^\ell\|_2^2$. Here, $\hat{\mathbf{e}}^\ell$ is obtained by decoding the top-$k_{\text{aux}}$ latent activations in $\mathbf{a}^\ell$ using the decoder matrix $\mathbf{W}_{\text{dec}}^{\ell \to \ell}$, where $k_{\text{aux}}$ is a hyperparameter. The full CLT training objective is then defined as

$$\mathcal{L}_{\text{CLT}} = \mathcal{L}_{\text{MSE}} + \alpha \mathcal{L}_{\text{aux}}, \tag{3}$$

where $\alpha$ controls the weight of the auxiliary loss. This joint objective enforces faithful reconstruction of MLP outputs while encouraging broad usage of the latent space, thereby increasing the number of interpretable latent features.

We train CLTs on ESM2-8M and ESM2-35M, for which $L = 6$ and $d_{\text{model}} = 320$ and $L = 12$ and $d_{\text{model}} = 480$, respectively. Following a procedure similar to (Adams et al., 2025), we train on 5 million protein sequences of length up to 1022 amino acids, randomly sampled from UniRef50 (Suzek et al., 2007). After training, the CLT functions as a replacement model for ESM2: at each layer,

it substitutes the original MLP block with a sparse, interpretable latent representation that approximates the original computation (Fig. 1a). For architectural and optimization details, see Appendix A. To enable controlled comparison, we also train PLTs using the same hyperparameters and sparsity constraint $k$ as their respective CLTs (see Appendix B).

## 2.2. Circuit Discovery

Leveraging the interpretable latent space induced by ProtoMech, we seek to recover the minimal subset of latent variables that governs task-specific computation in ESM2. While the CLT latent dimension is large, we posit that model behavior is mediated by a sparse collection of active latent features. Following (Dunefsky et al., 2024), we define a *circuit* as a set of latent variables whose joint activity is responsible for a specified model behavior.

**Replacement Model.** To identify such circuits, we first train a supervised probe on the final MLP output of ESM2, $\mathbf{y}^L$, to establish the performance of the original model on the target task. We then seek the minimal subset of ProtoMech latent variables capable of producing a reconstructed final MLP output, $\hat{\mathbf{y}}^L$, that recovers this performance (Fig. 1a).

To compute $\hat{\mathbf{y}}^L$, we adopt a replacement model strategy similar to that of (Ameisen et al., 2025). Specifically, during the forward pass, the original MLP blocks are replaced by the ProtoMech circuit, while the outputs of the attention heads are held fixed to those of the ground-truth ESM2 model. This hybrid formulation isolates the contribution of the MLP pathway while avoiding error accumulation from repeatedly reconstructing attention representations.

We observe that a fully recursive replacement, where both attention and MLP computations are derived from ProtoMech reconstructions at each layer, leads to substantial perfor-

mance degradation due to compounding reconstruction errors (Ameisen et al., 2025). A comprehensive comparison of alternative replacement strategies, including prior approaches and new variants introduced here, is provided in Appendix C. We apply this replacement model framework to two downstream tasks: protein family classification and protein function prediction.

**Protein Family Classification.** We assess ProtoMech on a collection of binary classification problems corresponding to protein family membership. We construct these tasks using Swiss-Prot sequences clustered at 30% sequence identity and annotated with InterPro family labels (Paysan-Lafosse et al., 2022). For each family, we train a logistic regression probe on $\mathbf{y}^L$, which serves as the reference computation that the ProtoMech circuit must reproduce.

**Function Prediction.** We next evaluate ProtoMech's ability to model protein fitness landscapes. We select 12 Deep Mutational Scanning (DMS) assays from ProteinGym (Notin et al., 2023a), spanning a diverse set of biological functions (see Table 2 for the full list of assays). For each DMS assay, we adopt the supervised cross-validation protocols provided by ProteinGym, which define two evaluation regimes: *(i)* training and testing exclusively on single mutants, and *(ii)* training and testing on both single and higher-order mutants. To establish the performance of the original model, we train a convolutional neural network (CNN) probe on the final-layer MLP output $\mathbf{y}^L$, following the training procedure described in (Notin et al., 2023b).

**Circuit Discovery Algorithm.** Similar to (Nainani et al., 2025; Dunefsky et al., 2024), to recover the minimal subset of latent variables required for each task, we employ an iterative greedy search procedure based on gradient-based attribution. For each latent unit, we compute an attribution score by measuring its contribution to the probe's output on a held-out validation set. Latents are then ranked by attribution magnitude and incrementally added to the candidate circuit in small batches. This procedure continues until the resulting circuit recovers at least 70% of the original ESM2 performance, or until it reaches the performance of the full replacement model when all latents are included. We use the F1 score as the evaluation metric for protein family classification and the Spearman rank correlation for function prediction. Additional details of the circuit discovery procedure are provided in Appendices D.1 and D.2.

### 2.3. Steering

To further validate the computational pathways discovered by ProtoMech, we intervene on the function prediction circuits to steer ESM2 toward high-fitness sequences (Fig. 2b). We focused our analysis on seven of the original 12 DMS assays, selected based on the availability of sufficient mutant data. To independently evaluate the fitness of the generated

variants, we trained a CNN evaluation model on 90% of the training data, utilizing the same architecture and training procedure as our circuit discovery CNN probes.

Our steering strategy involves activation clamping within the CLT replacement model, similar to (Templeton et al., 2024). During the forward pass of a wildtype sequence, we select a subset of latents within the target function circuit and fix their activations to a high activation. We choose this high activation by identifying the maximum activation magnitude observed for that node across the entire sequence, and multiply it by a scalar multiplier. We then utilize Equation (2), setting $\ell = L$, to obtain $\hat{\mathbf{y}}^L$ and then decode the resulting representation to ESM2's logits. We select mutations corresponding to the maximum probability values. To ensure the CNN evaluation model remains a valid proxy for experimental fitness, we constrained the generation to variants within a radius of five mutations from the wildtype. We note that steering via Equation (2) and setting $\ell = L$ is one of many possible ways, and we provide a comprehensive overview of all the steering methods tested in Appendix D.3. We report our results steering ProtoMech and PLT circuits following an ablation study conducted in Appendix E.

### 2.4. Visualizing Circuits

To qualitatively analyze the computational pathways discovered, we construct sparse graph representations using the ProtoMech visualizer (Fig. 2c), where nodes correspond to latents and edges represent the influence of the nodes on each other. For a given input sequence, we isolate the graph by selecting the top-five highest activating nodes per layer. When analyzing the mutational effects of sequences in function prediction circuits, we additionally include the top-five nodes exhibiting the largest activation shift between the wildtype and variant.

From here, we aim to identify possible biological overlap learned by each of the nodes. We identify the specific amino acids that are being activated in the input sequence and cross-reference them with the top-10 maximally activating sequences from Swiss-Prot to detect conserved motifs or family-specific patterns (Fig. 1b). We further project these activated amino acids onto the protein structure to visualize their spatial clustering and infer their biological relevance. Finally, to trace the computational path learned by ESM2, we connect our graph with edges following (Ameisen et al., 2025). Briefly, we compute the edge weight connecting a source node to a target node, where the target node must reside at a subsequent layer. This weight is calculated as the product of the source node's activation and the gradient of the target's pre-activation with respect to the source node (see Appendix F for more details).

# 3. Results

We evaluate the performance of ProtoMech on three distinct tasks: *(i)* circuit discovery, *(ii)* steering, and *(iii)* uncovering overlapping biological motifs in circuits. In the main text, we report results from ESM2-8M. For tests of other replacement models described in Appendix C and our steering ablation study, we refer to Appendix E. For results on scaling ProtoMech to ESM2-35M, we refer to Appendix G.

## 3.1. ProtoMech Compresses ESM2's Computation

**Protein Family Circuits.** We first evaluated the accuracy of ProtoMech on protein family classification tasks (Fig. 1c). Using the full set of latents, ProtoMech achieves an average F1 score of $0.82 \pm 0.19$, recovering approximately 89% of the original model's performance ($0.92 \pm 0.10$). This significantly outperforms the PLT baseline, which achieved an average F1 of $0.50 \pm 0.34$, suggesting that the underlying representation learned by ProtoMech provides a substantially more accurate approximation of ESM2's computation.

This performance advantage extends to circuit discovery. The circuits identified by ProtoMech achieve an average F1 of $0.73 \pm 0.19$, (79% recovery compared to the original model), compared to $0.49 \pm 0.33$ from the PLT. Notably, ProtoMech achieves this while using only $150 \pm 204$ latents on average (roughly 0.8% of the total possible latent space), demonstrating its ability to compress biologically relevant information into a sparse set of latents.

**Function Prediction Circuits.** We observe similar trends when applying ProtoMech to fitness regression tasks (Fig. 1d). On the full set of latents, ProtoMech achieves an average Spearman correlation of $0.41 \pm 0.19$, recovering approximately 82% of the original model's performance ($0.50 \pm 0.22$). ProtoMech again outperforms the PLT baseline, which achieved an average Spearman of $0.38 \pm 0.18$.

When identifying circuits, ProtoMech retains its advantage, achieving an average Spearman of $0.38 \pm 0.18$ (76% recovery) compared to $0.35 \pm 0.19$ for the PLT. ProtoMech again achieves this recovery while using only a tiny fraction of the possible latent space ($167 \pm 221$ latents, making up 0.9% of the total latent space).

To assess whether the function circuits discovered represent global functional motifs, we analyzed the GFP_AEQVI_Sarkisyan (Sarkisyan et al., 2016) DMS assay, which contains variants up to 10 mutations away. We then stratified the Spearman correlation by mutation depth between ESM2, ProtoMech with all latents, and the circuit discovered (Appendix E.1). We observe that ProtoMech circuits maintain their predictive power across all mutation depths, with ProtoMech circuits recovering up to 74% of the predictive power of ESM2 at 5+ mutations. This suggests that ProtoMech captures computational mechanisms that

are representative of global functional motifs.

## 3.2. ProtoMech Designs High-fitness Sequences

We next evaluated whether the computational pathways identified by ProtoMech could be used to steer the model toward high-fitness variants. To benchmark our approach, we compared it with steering using PLT circuits, selecting random mutations, and Contrastive Activation Addition (CAA) (Huang et al., 2025), a popular method for steering dense pLMs without a sparse latent representation. Briefly, CAA identifies a global "concept vector" in ESM2's hidden dimension by computing the difference between the mean activations ($\mathbf{y}^L$) of high-fitness and low-fitness sequences; this vector is then injected into the residual stream during generation to bias the model's output (see Appendix D.4).

Table 1 presents our steering results on the seven selected DMS assays. Interestingly, we find that steering via circuits (both ProtoMech and PLT) consistently yields sequences with higher fitness than CAA, and that circuit-based methods outperform CAA and random in all metrics. We attribute the limitations of CAA to data scarcity and the mutation-locality of DMS assays. First, CAA relies on averaging activations to estimate a robust direction of improvement. However, DMS datasets are often relatively small (containing as few as 1,000 high-fitness sequences in some splits), which may be insufficient to estimate a concept vector. Secondly, these sequences are extremely localized around the wildtype, typically only one to two mutations away. Consequently, the concept vector is likely overfitting to this localized region and is unable to extrapolate effectively to high-fitness regions of the fitness landscape. In contrast, circuit-based methods leverage a sparse, biologically disentangled latent space. By identifying and clamping only the specific nodes that govern protein function, steering with circuits minimizes interference from irrelevant features, allowing for higher fidelity in targeted generation. When comparing both ProtoMech and PLT directly, we find that ProtoMech outperforms PLT in 71% of cases, which includes generating the single highest-fitness variant, as well as the highest top 10% and top 20% fitness variants. This suggests that ProtoMech captures functional dependencies more effectively than PLT and can generate a diverse pool of highly functional variants.

## 3.3. ProtoMech Reveals Known Biological Motifs

Given the performance of ProtoMech, a natural question arises: as we trace the computational pathways of ESM2, do we uncover known *biological* motifs? To this end, we perform a rigorous qualitative analysis on our circuits, focusing on three representative case studies: protein kinase and NADP+ binding domains (from protein family classification) and the GB1 fitness landscape (from function

*Table 1.* Steering results using ProtoMech, PLT, CAA, and selecting random mutations on seven DMS assays. All variants were constrained to a maximum of five mutations away from the wildtype.

| Method | DMS | Mean score ↑ | Max score ↑ | Top 10% score ↑ | Top 20% score ↑ |
|---|---|---|---|---|---|
| **ProtoMech** | SPG1_STRSG_Olson_2014 | 1.67 ± 0.67 | **3.24** | **3.00 ± 0.21** | **2.70 ± 0.34** |
| | HIS7_YEAST_Pokusaeva_2019 | **1.28 ± 0.09** | **1.42** | **1.41 ± 0.01** | **1.40 ± 0.02** |
| | GRB2_HUMAN_Faure_2021 | **-0.15 ± 0.07** | 0.05 | -0.01 ± 0.04 | -0.05 ± 0.04 |
| | GFP_AEQVI_Sarkisyan_2016 | 4.17 ± 0.23 | 4.63 | 4.56 ± 0.05 | 4.50 ± 0.08 |
| | CAPSD_AAV2S_Sinai_2021 | **1.68 ± 1.08** | **4.54** | **3.77 ± 0.43** | **3.36 ± 0.53** |
| | RASK_HUMAN_Weng_2022_abundance | **-0.12 ± 0.06** | **0.06** | **0.02 ± 0.03** | **-0.02 ± 0.05** |
| | A4_HUMAN_Seuma_2022 | **0.12 ± 0.33** | **1.07** | **0.76 ± 0.18** | **0.63 ± 0.19** |
| **PLT** | SPG1_STRSG_Olson_2014 | **1.97 ± 0.48** | 3.18 | 2.88 ± 0.20 | 2.67 ± 0.25 |
| | HIS7_YEAST_Pokusaeva_2019 | 1.27 ± 0.06 | 1.41 | 1.38 ± 0.02 | 1.36 ± 0.03 |
| | GRB2_HUMAN_Faure_2021 | -0.18 ± 0.14 | **0.15** | **0.12 ± 0.05** | **0.03 ± 0.09** |
| | GFP_AEQVI_Sarkisyan_2016 | **4.40 ± 0.22** | **5.15** | **4.92 ± 0.19** | **4.76 ± 0.21** |
| | CAPSD_AAV2S_Sinai_2021 | 0.81 ± 0.53 | 2.05 | 1.87 ± 0.11 | 1.67 ± 0.22 |
| | RASK_HUMAN_Weng_2022_abundance | -0.19 ± 0.08 | **0.06** | -0.02 ± 0.06 | -0.07 ± 0.07 |
| | A4_HUMAN_Seuma_2022 | -0.05 ± 0.38 | 1.03 | 0.67 ± 0.24 | 0.52 ± 0.24 |
| **CAA** | SPG1_STRSG_Olson_2014 | 0.70 ± 0.00 | 0.70 | 0.70 ± 0.00 | 0.70 ± 0.00 |
| | HIS7_YEAST_Pokusaeva_2019 | 0.52 ± 0.14 | 0.77 | 0.77 ± 0.00 | 0.71 ± 0.06 |
| | GRB2_HUMAN_Faure_2021 | -0.40 ± 0.11 | -0.20 | -0.21 ± 0.02 | -0.23 ± 0.03 |
| | GFP_AEQVI_Sarkisyan_2016 | 2.93 ± 0.23 | 3.16 | 3.16 ± 0.00 | 3.16 ± 0.00 |
| | CAPSD_AAV2S_Sinai_2021 | -0.26 ± 0.55 | 0.45 | 0.45 ± 0.00 | 0.45 ± 0.00 |
| | RASK_HUMAN_Weng_2022_abundance | -0.35 ± 0.07 | -0.28 | -0.28 ± 0.00 | -0.29 ± 0.01 |
| | A4_HUMAN_Seuma_2022 | -1.90 ± 0.03 | -1.87 | -1.87 ± 0.00 | -1.87 ± 0.00 |
| **Random** | SPG1_STRSG_Olson_2014 | -2.76 ± 0.93 | -1.25 | -1.37 ± 0.09 | -1.56 ± 0.22 |
| | HIS7_YEAST_Pokusaeva_2019 | 0.56 ± 0.25 | 1.08 | 0.93 ± 0.07 | 0.88 ± 0.07 |
| | GRB2_HUMAN_Faure_2021 | -0.84 ± 0.26 | -0.37 | -0.45 ± 0.06 | -0.50 ± 0.07 |
| | GFP_AEQVI_Sarkisyan_2016 | 2.74 ± 0.51 | 3.54 | 3.46 ± 0.05 | 3.35 ± 0.12 |
| | CAPSD_AAV2S_Sinai_2021 | -1.04 ± 0.61 | 0.50 | 0.09 ± 0.39 | -0.26 ± 0.45 |
| | RASK_HUMAN_Weng_2022_abundance | -0.64 ± 0.26 | -0.04 | -0.21 ± 0.09 | -0.27 ± 0.09 |
| | A4_HUMAN_Seuma_2022 | -1.57 ± 0.61 | -0.87 | -0.94 ± 0.05 | -0.99 ± 0.07 |

prediction). Our analysis reveals that ProtoMech successfully disentangles ESM2 into interpretable computational pathways that overlap with known biological motifs.[2]

**Protein Kinase Domain.** Fig. 3a depicts a circuit discovered governing the protein kinase domain (InterPro ID IPR000719), which catalyzes the chemical reactions that regulate cell signaling and behavior (Taylor & Kornev, 2011). We observe that earlier layers tend to recognize specific amino acids that make up more complex motifs in later layers. Specifically, layer 1 at latent 1582 (L1/1582) activates on arginine (R) amino acids, which serve as essential building blocks for catalytic activity (Kornev et al., 2006). This feeds directly into layer 3 (L3/1918), which identifies the HRD motif—a highly conserved catalytic loop responsible for binding to substrate peptides for phosphorylation, the key mechanism for kinase signaling (Modi & Dunbrack, 2019). Subsequently, layer 5 is split into different motifs: L5/1774 identifies the ATP-binding site, a pocket that facilitates the docking of ATP for phosphorylation (Taylor & Kornev, 2011), while L5/826 detects the Glycine-rich

loop (G-loop), which anchors ATP during catalysis (Bossemeyer, 1994). Consistent with findings in (Adams et al., 2025), we observe that the final layer also tends to recognize specific amino acids. For instance, layer 6 (L6/886) activates strongly on serine (S) amino acids. One possible explanation is that, because the representative input sequence used to model these activations belongs to the serine/threonine-protein kinase family, ESM2 aggregates structural context learned in earlier layers to pinpoint the specific serine residue required for phosphorylation.

**NADP+ Binding Domain.** Fig. 3b depicts a circuit governing the NADP+ binding domain superfamily (InterPro ID IPR036291). In this particular example, we focus on the protein Metalloreductase STEAP3, which binds to Nicotinamide Adenine Dinucleotide Phosphate (NADP+) and Flavin Adenine Dinucleotide (FAD) in order to facilitate the electron transfer of iron for the cell (Ohgami et al., 2005). We similarly find that earlier layers tend to recognize specific amino acids. For instance, L1/967 activates on phenylalanine (F) residues, which are present in the NADP+ binding sites. This latent variable feeds into layer 4 (L4/53), which identifies the Rossmann fold, a structural motif composed of alternating beta strands and alpha helices that facilitate NADP+ binding (Hanukoglu, 2015). From here, this

---

[2]Figs. 3 and 4 display a subset of findings selected for annotation availability and representative cross-layer coverage. These circuits aim to provide an overview of the mechanisms captured by ProtoMech and are by no means exhaustive.

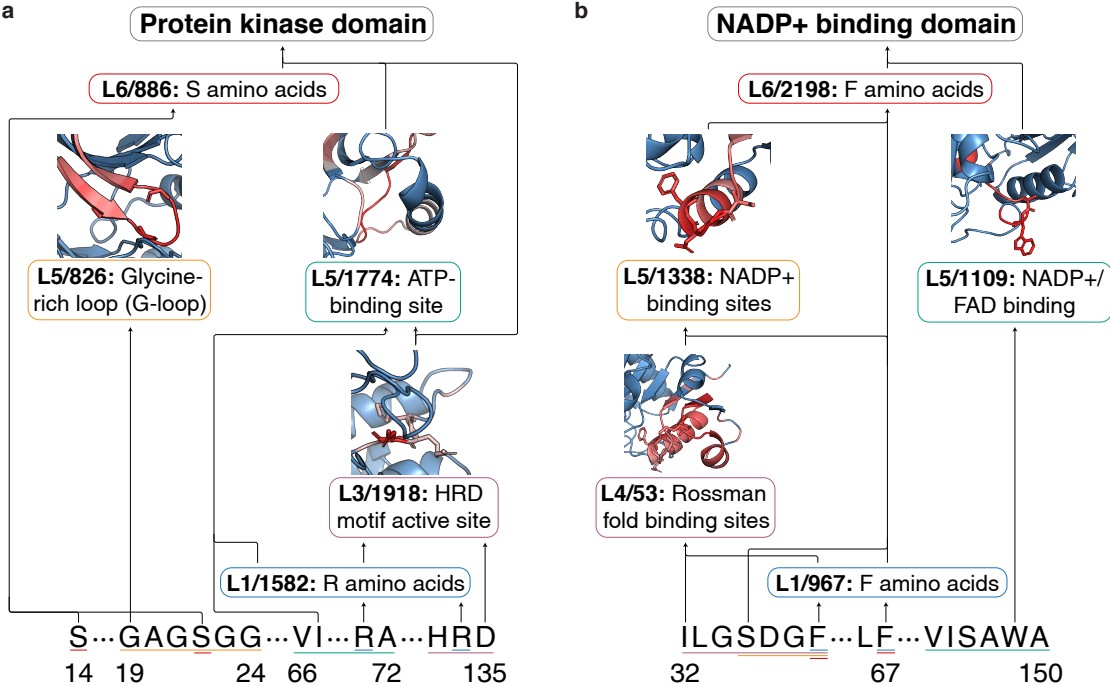

*Figure 3.* **Examples of family circuits discovered using ProtoMech.** We use the ProtoMech visualization tool to examine **a**, protein kinase domain and **b**, NADP+ binding domain circuits. We find interpretable features related to binding and active sites, secondary structure, and biochemical patterns. We observe that earlier layers are detecting key amino acids that assemble into complex motifs.

information is fed into layer 5 (L5/1338), which narrows this context into specific NADP+ binding pockets within the Rossmann fold. Separately, layer 5 (L5/1109) also detects NADP+ and FAD-binding sites in the sequence. Finally, in the last layer of ESM2, we observe repeated concepts from earlier layers. For instance, layer 6 (L6/2198) reactivates F amino acids as in layer 1. This recurrence mirrors mechanistic interpretability findings in natural language (Ameisen et al., 2025), suggesting that ESM2 reasserts specific residue details in its predictions.

**Mutation Sensitivity in GB1.** We evaluated the mutational sensitivity of our discovered circuits toward high- and low-fitness variants, focusing our analysis on the IgG-binding domain of Protein G (GB1). GB1 is a streptococcal surface protein that functions by binding to the Fc fragment of immunoglobulin G (IgG) antibodies (Sloan & Hellinga, 1999). When inputting the wildtype sequence into ProtoMech (Fig. 4a), we found that layer 1 (L1/249) activates on tryptophan (W) at position 43 (W43). W43 is a critical aromatic residue buried in GB1's hydrophobic core, which stabilizes the protein and binds to Fc (Sauer-Eriksson et al., 1995). This feeds into layer 3 (L3/3027), which detects hydrophobic interactions, most notably between leucine (L) at position 5 and W43. The interaction between L5 and W43 is well-documented in literature, and is known to contribute to GB1's high stability (Gronenborn et al., 1991). Separately, L1/249 also feeds into layer 6 (L6/1610), which similarly

detects W amino acids, mirroring our findings from the NADP+ binding domain with ESM2 learning repeating concepts.

We then leveraged the DMS assay from (Olson et al., 2014) to investigate the mechanistic drivers of high-fitness variants. To this end, we analyzed a known high-fitness variant, A24W (alanine to tryptophan at position 24) (Fig. 4b). Interestingly, while all the latents that made up the circuit from the wildtype remain active, ProtoMech identifies a feature previously not seen in the wildtype, L6/3029, which activates on inward-facing and hydrophobic residues. This suggests that A24W, by introducing a bulky aromatic amino acid, packs more efficiently into the protein's hydrophobic core, improving stability and therefore binding affinity.

Conversely, we also analyzed a known low-fitness variant, W43Q (Tryptophan to Glutamine at position 43) (Fig. 4c). Given that W43 is a critical interface residue, we hypothesized that the circuit would reflect a loss of both binding affinity and stability. Indeed, feeding the mutated sequence into ProtoMech reveals the deactivation of the W amino acid detectors L1/249 and L6/1610, confirming the loss of a binding site. Crucially, the mutation also weakens L6/3027, which previously activated hydrophobic interactions. This suggests that W43Q destabilizes GB1 and destroys the Fc-binding site, thereby preventing GB1 from performing its function. These examples underscore ProtoMech's ability

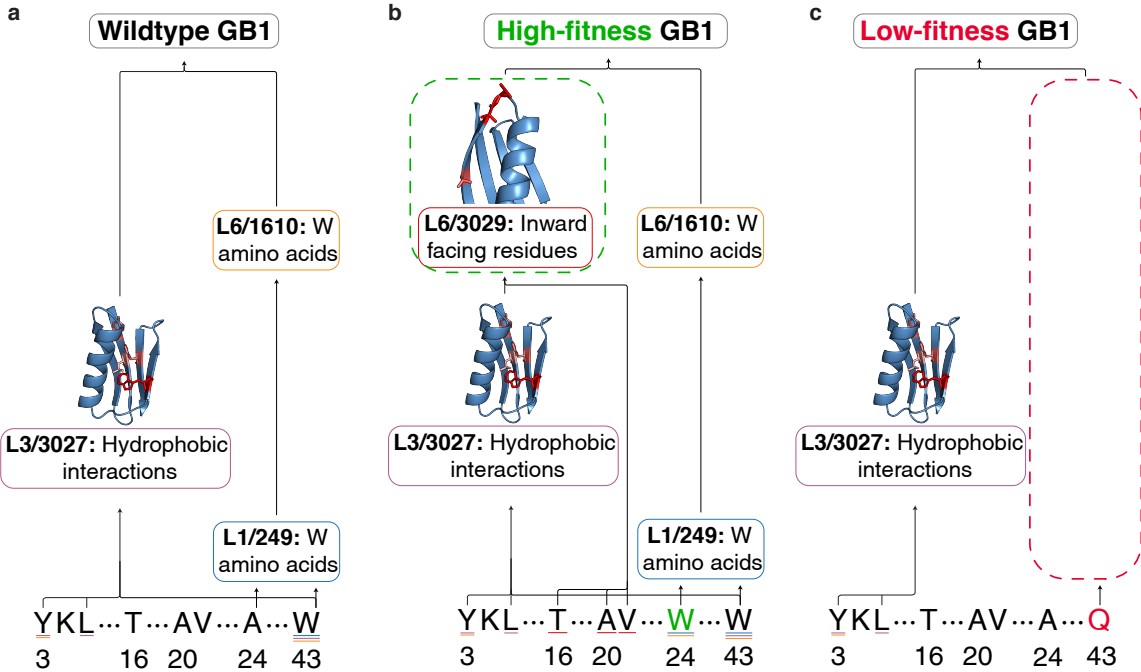

*Figure 4.* **Examples of the mutation sensitivity of function circuits discovered using ProtoMech. a**, We feed in the wildtype sequence of GB1 into ProtoMech and identify latents related to binding affinity and stability. **b**, A high-fitness variant of GB1 activates an additional latent that corresponds to protein stability. **c**, Conversely, a low-fitness variant of GB1 deactivates parts of the circuit related to binding affinity and stability. These findings highlight ProtoMech's ability to provide a mechanistic rationale for changes in fitness to sequences.

to detect high- and low-fitness mutations and to provide interpretable, biologically plausible explanations.

## 4. Discussion

**Contrasting Sparse Autoencoders.** ProtoMech fundamentally differs from SAE-based approaches to interpreting pLMs. While SAEs focus on decomposing a single activation state (see Appendix H for more details), ProtoMech functions as a replacement model by explicitly learning the computation going from one representation to the next. Although recent work has utilized pretrained SAEs to infer circuit connectivity in pLMs (Nainani et al., 2025), such methods are still reliant on decomposing single activation states and thus cannot serve as a replacement model.

**Denoising ESM2.** Surprisingly, we observe that our discovered circuits occasionally *outperform* the performance of the full ESM2 model on both protein family classification and function prediction. This phenomenon is most pronounced when ESM2 struggles. For instance, in protein family classification tasks where ESM2 achieves an F1 score below 0.5 ($0.39 \pm 0.04$), ProtoMech circuits achieve a higher average performance ($0.43 \pm 0.27$) (Fig. 8). In fitness prediction tasks, although ESM2 retains a higher Spearman correlation on average, 37% of our circuits achieve a higher performance. We attribute this to a *denoising* (regularization) effect inherent to ProtoMech's sparse latent representation.

By transcoding ESM2 through a sparse latent space, ProtoMech effectively filters out task-irrelevant noise, distilling the computation down to ESM2's most salient features. This suggests that ProtoMech offers utility beyond interpretability, potentially as a regularizer to enhance pLM robustness.

**High-Throughput Protein Screening.** Across our case studies, we demonstrate that ProtoMech captures known biological motifs and distinguishes between high- and low-fitness mutations. One potential application of ProtoMech in this regard is high-throughput protein screening, which necessitates generating and screening thousands of sequences to create large-scale functional libraries (Wu et al., 2019; Walton et al., 2025b). In this context, ProtoMech offers a solution as a mechanistic filter. By tracing computational pathways, researchers can better identify variants utilizing biologically plausible motifs, enabling the selection of higher-quality candidates for wet-lab synthesis.

**Extending ProtoMech to Other pLM Architectures.** In this study, we validate ProtoMech using ESM2, a masked language model. As ESM2 is arguably the most widely used model for protein function prediction and fitness modeling, it is imperative that any mechanistic interpretability tool first demonstrate success here. However, we acknowledge the growing diversity of pLM architectures, including autoregressive and diffusion-based models. Extending ProtoMech to these other architectures is not trivial. Autoregressive

models often require alignment or fine-tuning strategies for effective generation, which fundamentally differ from the supervised probing we utilize for fitness extrapolation. Furthermore, to the best of our knowledge, CLTs have not yet been applied to diffusion models. Therefore, we consider both these extensions to be beyond the scope of our current study and remain open challenges.

**Scaling ProtoMech.** One difficulty with scaling ProtoMech is the number of parameters required to train a CLT. Due to the CLT's dependence on cross-layer connections, the number of decoder matrices required scales with $\mathcal{O}(L^2)$, compared to $\mathcal{O}(L)$ with the number of encoder matrices. In practice, training a CLT on ESM2-8M required approximately 28M parameters, a $3.5\times$ increase over the original model's size.

To evaluate the computational feasibility of CLTs on larger pLMs, we scaled up ProtoMech to ESM2-35M, mirroring the family and function circuit discovery tasks performed on the ESM2-8M model (Appendices G.1 and G.2) We observe similar trends to the results on ESM2-8M. Notably, ProtoMech circuis were able to recover up to 85% of the performance of ESM2-35M, compared to up to 79% of the performance of ESM2-8M, suggesting that scaling CLTs may also be increasingly effective at capturing the computational mechanisms in state-of-the-art pLMs.

Secondly, to address the scalability of CLTS in larger pLMs, we propose "windowed" CLTs, a strategy inspired by (Shu et al., 2026) (see Appendix G.3 for more details). Unlike vanilla CLTs, which require global connectivity across all preceding layers, windowed CLTs approximate cross-layer connectivity in localized windows. For instance, in ESM2-35M, which has 12 layers, instead of mandating that the reconstruction of $\mathbf{y}^{12}$ depends on all 12 layers, we can constrain it to a localized window of its four preceding layers. To validate this architecture, we trained a windowed CLT on ESM2-35M, which reduced our parameter count by 40% (207M to 125M) and sped up training by $1.75\times$ (see Table 10 for estimated training times across all ESM2 models). On family discovery tasks, windowed CLTS captured 82% of ESM2-35M's performance, surpassing the PLT baseline (68%) while approximating the vanilla CLT's performance (85%). These results demonstrate that windowed CLTs strike a fair tradeoff between capturing cross-layer dependencies and reducing compute time.

**Limitations.** One limitation lies in the interpretation of ProtoMech circuits, which currently relies on manual analysis coupled with existing biological annotations. As our capacity to label circuits is bounded by current biological knowledge, it is possible that circuits governing mechanisms are not yet well-characterized. Furthermore, our reliance on human interpretation to label our circuits restricts the scale of our qualitative analysis. Developing automated annota-

tion pipelines remains a critical next step to accelerate the discovery of novel biological insights.

**Conclusion.** In this paper, we introduce ProtoMech, a framework for tracing computational circuits in protein language models via cross-layer transcoders. We demonstrate that these circuits are highly compressible and steerable toward high-fitness sequences, while consistently overlapping with known biological motifs. Our work opens the door toward principled circuit tracing in pLMs.

## Impact Statement

This paper presents work whose goal is to advance the field of Machine Learning, specifically toward interpreting protein language models. The potential societal impacts of our work are overwhelmingly positive. Potential applications of this work include discovering biological mechanisms, enhancing protein design, and improving the performance of downstream tasks.

## Acknowledgment

This research was supported by the National Science Foundation (NSF) Graduate Research Fellowship Program (GRFP), the Parker H. Petit Institute for Bioengineering and Biosciences (IBB) interdisciplinary seed grant, the Exponential Electronics seed grant of the Institute for Matter and Systems (IMS) at Georgia Tech, Microsoft via GT Cloud Hub, Georgia Tech Undergraduate Research Opportunities Program, Georgia Tech Research Corporation, and Georgia Institute of Technology start-up funds.

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

## A. Model Architecture and Hyperparameters

On ESM2-8M, where $L = 6$ and $d_{\text{model}} = 320$, we train our models on 5 million random sequences under 1022 residues from UniRef50. Before mapping each layer's residual stream activations into their respective latent spaces, we first apply the TopK function. For the CLT and PLT, we set $k = 16$ at each layer to achieve a total sparsity of $16 \times 6 = 96$ latents, which is consistent with the level of sparsity targeted by (Ameisen et al., 2025). We additionally set $d_{\text{latent}} = 3200$, which is a $10\times$ expansion factor from the hidden dimension of ESM2-8M. We loosely take inspiration from (Simon & Zou, 2025; Adams et al., 2025) on the choice of expansion factor, balancing computational efficiency and a large enough latent space to capture overlapping biological motifs. Taking inspiration from (Gao et al., 2025), we additionally set $k_{\text{aux}} = 32$ and $\alpha = 1/32$. We note that the auxk loss's primary objective is to mitigate dead latents by forcing them to activate. For computational efficiency, we choose to activate them using the decoder matrix $\mathbf{W}_{\text{dec}}^{\ell \to \ell}$. On ESM2-35M, where $L = 12$ and $d_{\text{model}} = 480$, we set $k = 24$ and $d_{\text{latent}} = 4800$. Additionally, in ESM2-8M, we apply an explicit LayerNorm operation to the input of the MLP at every layer, which is common in some sparse autoencoder architectures. For ESM2-35M, we removed this step since the base ESM2 architecture already incorporates internal normalization for its MLP inputs. All other hyperparameters we keep the same as in ESM2-8M.

We train all our models with a batch size of 16 and a learning rate of $2 \times 10^{-4}$ with the Adam optimizer over 250,000 steps. To help stabilize training, we utilize a gradient clipping value of 1 and weight decay of value $1 \times 10^{-5}$. We additionally normalize the mean-squared error at each layer by dividing by the variance of the ground-truth residual stream activations at each batch. With our choice of hyperparameters, we obtain an average normalized mean-squared error of $0.15$ and $0.13$ on a held-out validation set during training on ESM2-8M and ESM2-35M, respectively.

We note that a popular method for constraining the sparsity of CLT activations is using a $\tanh$ loss (Ameisen et al., 2025). While an interesting future direction, this adds significant computational burden, as it requires extensive hyperparameter tuning to balance the regularizer's strength across different architectures. Furthermore, a $\tanh$ loss would result in varying sparsity levels between models, which would complicate experiments and hinder direct comparisons between the CLT and PLT. Adopting TopK in this paper allows us to bypass these complexities and set up a controlled environment for model evaluation.

## B. Per-layer Transcoders

To benchmark our CLT, we train a per-layer transcoder (PLT), where each layer has a single transcoder trained independently of each other layer (Ameisen et al., 2025; Dunefsky et al., 2024). Here, each layer's residual stream activations $\mathbf{x}^\ell$ are mapped to their respective $\mathbf{a}^\ell$ via:

$$\mathbf{a}^\ell = \text{TopK}(\mathbf{W}_{\text{enc}}^\ell(\mathbf{x}^\ell - \mathbf{b}_{\text{pre}}^\ell) + \mathbf{b}_{\text{enc}}^\ell). \tag{4}$$

Mathematically, the PLT encoder mapping is identical to the CLT encoder mapping from Equation (1). However, computing $\hat{\mathbf{y}}$ now uses a single decoder which only has access to the information found in layer $\ell$:

$$\hat{\mathbf{y}}^\ell = \mathbf{W}_{\text{dec}}^{\ell \to \ell} \mathbf{a}^\ell + \mathbf{b}_{\text{pre}}^\ell. \tag{5}$$

For consistency in notation, we denote $\mathbf{W}_{\text{dec}}^{\ell \to \ell}$ as the PLT's decoder at layer $\ell$, but note that for each layer, there exists only a single decoder matrix unlike in a CLT.

The total training objective, $\mathcal{L}_{\text{PLT}}$, is similar to the CLT training objective and is defined as $\mathcal{L}_{\text{PLT}} = \mathcal{L}_{\text{MSE}} + \alpha \mathcal{L}_{\text{aux}}$. Since the PLT does not have access to cross-layer decoders, our setup is identical to training $L$-many independent transcoders.

## C. Replacement Models

In this section, we describe how we design replacement models to approximate $\mathbf{y}^L$ using both CLTs and PLTs. First, we denote transformer notation to the describe the forward pass of ESM2. Then, we go over the ProtoMech and PLT replacement models.

### C.1. Transformer Notation

Following a similar notation established in (Elhage et al., 2021; Dunefsky et al., 2024), we represent the forward pass of a transformer as follows. Let $\mathbf{x}_{\text{pre}}^\ell$ denote the residual stream activations at the beginning of layer $\ell$ before applying attention.

First, the attention sublayer computes its output based on $\mathbf{x}_{\text{pre}}^{\ell}$ and adds it to the residual stream to produce $\mathbf{x}^{\ell}$:

$$\mathbf{x}^{\ell} = \mathbf{x}_{\text{pre}}^{\ell} + \sum_{h \in H_{\ell}} h(\mathbf{x}_{\text{pre}}^{\ell}), \tag{6}$$

where $H_{\ell}$ represents the set of attention heads at layer $\ell$ and $h(\cdot)$ denotes the function computed by head $h$. Next, the MLP layer $\text{MLP}^{\ell}(\cdot)$ takes in $\mathbf{x}^{\ell}$ to produce $\mathbf{y}^{\ell}$, which is the target our CLTs are trained to reconstruct:

$$\mathbf{y}^{\ell} = \text{MLP}^{\ell}(\mathbf{x}^{\ell}). \tag{7}$$

Finally, $\mathbf{y}^{\ell}$ is added back to the residual stream to produce $\mathbf{x}_{\text{pre}}^{\ell+1}$:

$$\mathbf{x}_{\text{pre}}^{\ell+1} = \mathbf{x}^{\ell} + \mathbf{y}^{\ell}. \tag{8}$$

## C.2. ProtoMech Replacement Models

In the ProtoMech, we test the following replacement models for circuit discovery: direct, sequential, and full replacement. As reconstruction errors can compound when replicating the forward pass of ESM2 (Ameisen et al., 2025), these replacement models differ in how they combat the compounding reconstruction error. In the main text, we compare the performance of ProtoMech and PLT using the *sequential* replacement model on circuit discovery tasks to ensure we can compare the representational power of both architectures fairly. However, for the steering results in the main text, we utilize the *direct* replacement model in ProtoMech (keeping the sequential replacement model for the PLT), as our primary concern is designing the highest-functioning variants. For a full comparison of results across all replacement models, we refer to Appendix E.

**Direct.** The direct model approximates $\mathbf{y}^{L}$ directly without going through the forward pass. Here, we assume we have access to all ground-truth residual stream activations $\mathbf{x}^{\ell}$ at each layer. Based on Equation (2), to compute $\hat{\mathbf{y}}^{L}$, we then need to compute the summation over all decoders from that map from $\ell' = 1, 2, \ldots, L$. This method is only valid for CLTs (and not PLTs) because ablating latents in the CLT will change the output of the last layer even with ground-truth residual stream activations, while it will not change with the PLT.

**Sequential.** The sequential model does not assume access to all ground-truth residual stream activations $\mathbf{x}^{\ell}$. Here, we assume we have access to only $\mathbf{x}_{\text{pre}}^{0}$ as well as the ground-truth attention computations at each layer $h \in H_{\ell}$. Given this, the sequential model aims to replicate the entire forward pass through ESM. Given $\mathbf{x}_{\text{pre}}^{0}$, $\mathbf{x}^{0}$ is computed via Equation (6). The replacement model then approximates $\mathbf{y}^{0}$ as $\hat{\mathbf{y}}^{0}$ and, following Equation (8), produces the approximation of the next residual stream as $\hat{\mathbf{x}}_{\text{pre}}^{1} = \mathbf{x}^{0} + \hat{\mathbf{y}}^{0}$. From here, to obtain an approximation of $\hat{\mathbf{x}}^{1}$, we assume ground-truth access to $h(\mathbf{x}_{\text{pre}}^{1})$, and proceed the same as before. We repeat the process until reaching $\hat{\mathbf{y}}^{L}$.

**Full Replacement.** Using terminology from (Merullo et al., 2025), we define the *full replacement model* as the model that replaces $\mathbf{y}^{\ell}$ with its CLT reconstruction at each layer $\ell$. The full replacement model is similar to the sequential model, except it does not assume access to anything except $\mathbf{x}^{0}$. Given $\mathbf{x}_{\text{pre}}^{0}$, we compute $\hat{\mathbf{x}}_{\text{pre}}^{1}$ as in the sequential model. However, to obtain $\hat{\mathbf{x}}^{1}$, we do not assume access to $h(\mathbf{x}_{\text{pre}}^{1})$ and instead utilize $h(\hat{\mathbf{x}}_{\text{pre}}^{1})$. We repeat the process until reaching $\hat{\mathbf{y}}^{L}$.

Note that for both the sequential and full replacement model, error propagates throughout the forward pass, as the activations are found via:

$$\hat{\mathbf{a}}^{\ell} = \text{TopK}(\mathbf{W}_{\text{enc}}^{\ell}(\hat{\mathbf{x}}^{\ell} - \mathbf{b}_{\text{pre}}^{\ell}) + \mathbf{b}_{\text{enc}}^{\ell}),$$

which occurs at every layer. Conversely, the direct model bypasses this and reconstructs $\mathbf{a}^{\ell}$ by passing the ground truth value of $\mathbf{x}^{\ell}$ at each layer through the encoder.

## C.3. PLT Replacement Models

In the PLT, we test the sequential and full replacement models for circuit discovery.

**Sequential.** The sequential model is the analog to the ProtoMech sequential model. The only values we have access to are $\mathbf{x}_{\text{pre}}^{0}$ and the ground-truth attention outputs for each $h \in H_{\ell}$. Similar to the ProtoMech sequential model, the residual stream's attention update is done using the ground-truth attention values, while the MLP approximation $\hat{\mathbf{y}}^{\ell}$ is done using the PLT trained for layer $\ell$.

**Full Replacement.** Similarly to the ProtoMech full replacement model, the MLP of ESM replaced with the PLT trained at a given layer, and no information is given other than $\mathbf{x}_0$.

We note that no analog to the direct replacement model exists for the PLT. This is due to the lack of cross-layer connections: ablation at a layer $\ell' < \ell$ does not affect the reconstruction of $\hat{\mathbf{y}}^\ell$. Accordingly, the best way to measure the effect of ablating latents on the output of a PLT is to use a replacement model that compounds errors between sequential layers.

## D. Experimental Setup

In this section, we detail our experimental setup for circuit discovery and steering. The majority of our experimental setup is identical in ESM2-8M and ESM2-35M, with differences primarily arising due to computational constraints.

**Discovery with Supervised Models.** Throughout this paper, we use supervised models to anchor circuit discovery to biologically relevant tasks. While circuit discovery in natural language is typically anchored to next-token prediction (Ameisen et al., 2025), we argue that supervised tasks provide a more principled objective for ESM2. For one, ESM2 is a masked language model, meaning that it is not designed for next-token prediction like in autoregressive models. Additionally, pLMs like ESM2 are primarily utilized to encode global biological properties such as protein family or function. Consequently, anchoring circuits to next-amino acid predictions would fail to capture the structural and functional motifs that drive these attributes. We believe that the supervised tasks discussed in this paper are a representative subset of the areas circuit discovery would be utilized in pLMs.

### D.1. Protein Family Classification Circuit Discovery

**Data.** To identify family circuits, we use Swiss-Prot (Boeckmann et al., 2003) protein sequences obtained from (Adams et al., 2025). Here, Swiss-Prot protein sequences under 1022 residues were clustered at at 30% sequence identity using MMseqs2 (Steinegger & Söding, 2017). We collected and mean-pooled the MLP output from layer $L$ across all sequences in the dataset.

**Logistic Regression Probe Training.** We identified protein families by their InterPro ID (Paysan-Lafosse et al., 2022). For each protein family with 50 or more protein sequences, which we set to ensure we have enough data to discover a robust circuit, we assemble training data for the probe as the protein family sequences plus randomly sampled $4\times$ the number of sequences not from that protein family. We performed a stratified split to partition the data into training, validation, and test sets. We use 90% of the data for our training set. From the remaining 10% of data, we selected up to 128 sequences to form a validation set, which is reserved specifically for circuit discovery and computing attribution scores. We then trained a logistic regression probe for a maximum of 1000 iterations with the *balanced* class weight toggle using scikit-learn and computed all F1 scores based on the test set.

**Circuit Discovery.** Following a similar procedure to (Nainani et al., 2025), we quantitatively define a circuit as the smallest subset of latents required to achieve a comparable F1 score to the original logistic regression probe, $m_{clean}$. For each protein family, we define the comparable F1 score, $\theta$, to be 70% of the original F1 score. However, there may be cases where even adding all the latents do not recover 70% of the original F1 score, due to the nature of lossy reconstruction. To combat this, we compute the F1 score using CLT and PLT reconstructions with all latents active, $m_{all}$, and adjust the target to be the maximum possible performance achievable: $\theta = \min(0.7 \times m_{\text{clean}}, m_{\text{all}})$.

To identify the subset of latents, we employ an iterative greedy search based on gradient attribution. First, we compute an attribution score for every latent $i$ at every layer $\ell$, denoted as $\Delta_i^\ell$, with respect to the probe's prediction target $y$. We compute the attribution score as a product of the latent's activation $a_i^\ell$ and its gradient with respect to the prediction of $y$:

$$\Delta_i^\ell = \left| a_i^\ell \cdot \frac{\partial y}{\partial a_i^\ell} \right|. \tag{9}$$

$\Delta_i^\ell$ estimates the contribution of each latent to the model's output. We sum these attribution scores across all family sequences in the held-out validation set to obtain a global attribution ranking for each latent.

From here, we rank all latents by descending attribution score. We iteratively construct candidate circuits by adding the top-ranked latents in steps of 32 at a time, up to a maximum of 1000 latents. At each step, we compute the circuit performance $m_{\text{circuit}}$ using the sparse reconstruction derived from the top latents, and continue until $m_{\text{circuit}} \geq \theta$ or until we reach 1000 latents. We note that for computational efficiency, we opted to keep the validation set relatively small. However,

increasing the number of family sequences in the validation set will likely increase the quality of the circuit discovered.

**Plotting.** To create Fig. 1c and Fig. 5a, we only include the results of protein families with over 50 sequences, as to ensure the statistics we report are robust. These circuits are used to plot the distribution of latents in Fig. 6a. To create Fig. 8, we pull from all families regardless of size, but restrict the protein families where the F1 score is between 0.01 and 0.5. We specify that the F1 score should be above 0.01, as an F1 score of 0 means that ESM2 is not learning any information about the protein family. This means that circuit discovery will not distill any meaningful computations. We leave all protein family circuits on our HuggingFace.

### D.2. Function Prediction Circuit Discovery

*Table 2.* Summary of DMS assays used.

| DMS | Description | Function Tested | Single Mutants | Multiple Mutants |
|---|---|---|---|---|
| A4_HUMAN_Seuma (Seuma et al., 2022) | Amyloid-beta peptide | Aggregation | 796 | 14,015 |
| AMFR_HUMAN_Tsuboyama (Tsuboyama et al., 2023) | E3 ubiquitin-protein ligase AMFR | Stability | 820 | 2,152 |
| BBC1_YEAST_Tsuboyama (Tsuboyama et al., 2023) | Myosin tail region-interacting protein MTI1 | Stability | 1,084 | 985 |
| CAPSD_AAV2S_Sinai (Sinai et al., 2021) | Adeno-associated virus capsid | Viral production | 532 | 41,786 |
| DLG4_HUMAN_Faure (Faure et al., 2022) | Third PDZ domain of PSD95 | Yeast growth | 1,280 | 5,696 |
| F7YBW8_MESOW_Ding (Ding et al., 2024) | Antitoxin ParD3 | Growth enrichment | 80 | 7,842 |
| GFP_AEQVI_Sarkisyan (Sarkisyan et al., 2016) | Green fluorescent protein | Fluorescence | 1,084 | 50,630 |
| GRB2_HUMAN_Faure (Faure et al., 2022) | C-terminal SH3 domain of GRB2 | Yeast growth | 1,034 | 62,232 |
| HIS7_YEAST_Pokusaeva (Pokusaeva et al., 2019) | IGP dehydratase (HIS3) | Growth | 168 | 495,969 |
| RASK_HUMAN_Weng_abundance (Weng et al., 2023) | KRAS | Yeast growth | 3,066 | 22,946 |
| SPG1_STRSG_Olson (Olson et al., 2014) | IgG-binding domain of protein G | Binding | 1,045 | 535,917 |
| YAP1_HUMAN_Araya (Araya et al., 2012) | hYAP65 WW domain | Peptide binding | 362 | 9,713 |

**Data.** To identify function circuits, we use 12 DMS assays from ProteinGym (Notin et al., 2023a). We selected these proteins to ensure robust evaluation across a wide variety of functions (Table 2). Additionally, all of these DMS assays contain multiple mutants, which we utilize for our steering experiments. For each DMS, we collected their MLP outputs from layer $L$ without mean-pooling across all sequences.

**Training Splits**. We follow the supervised ProteinGym benchmark to determine our training data. The supervised benchmark is divided into two groupings: single and multiple mutations. In the single grouping, we train our CNN probes on single mutants and test on single mutants. ProteinGym uses three distinct cross-validation schemes to assess the ability of the CNN probe. In the *random* scheme, each mutation is randomly assigned to one of five folds. In the *contiguous* scheme, the sequence is split up into five continuous segments along its length. Mutations are assigned to each fold depending on which segment their position in the sequence falls under. Lastly, in the *modulo* scheme, the modulo operator is used to assign mutated positions to each fold. For example, position one is assigned to fold one, position two is assigned to fold two, continuing until position five is assigned to fold five. Position six is then looped back to fold one, position seven is looped back to fold two, and so on and so forth until the end of the sequence. In the multiple grouping, ProteinGym utilizes only the random scheme, since multiple mutations can exist outside the folds defined by the contiguous and modulo schemes. We train our CNN probes using five-fold cross validation, reporting the Spearman correlation each time. Three of the folds are reserved for training and use the validation fold to monitor loss for early stopping. Similar to the family circuit setup, the validation fold is also used for circuit discovery and computing attribution scores. We compute all Spearman correlations based on the test set. Due to computational constraints, in ESM2-35M for DMS assays SPG1_STRSG_Olson _2014, HIS7_YEAST_Pokusaeva _2019, GRB2_HUMAN_Faure_2021, and CAPSD_AAV2S_Sinai_2021, we randomly sample 1024 sequences from the test fold to use as our test set. Additionally for similar reasons, in we constrain the validation of all multiple mutation DMS assays on ESM2-35M, we randomly sample 128 sequences from the validation fold for our greedy circuit search algorithm, where half the sequences come from randomly sampling sequences above the median DMS score and the other half from randomly sampling sequences below the median DMS score.

**CNN Probe Training.** Following a similar methodology to (Notin et al., 2023b), the CNN architecture is set as a 1D convolutional layer with a kernel size of 7 and same padding, followed by a ReLU activation and dropout with a probability of 0.1. The output is then passed through a linear head to produce a scalar fitness prediction. We train the probe using the AdamW optimizer with a maximum learning rate of $3 \times 10^{-4}$ and a cosine annealing schedule with a linear warmup of 100 steps. Training proceeds for 10,000 steps with a batch size of 128, minimizing the Mean Squared Error (MSE) loss. We employ early stopping with a patience of 10 evaluation steps based on the validation loss.

**Circuit Discovery.** Similar to the family circuit setup, we define $m_{\text{clean}}$ to be the Spearman correlation of the original CNN

probe. We then compute $m_{\text{all}}$ as the Spearman correlation with all the latents active, and set $\theta = \min(0.7 \times m_{\text{clean}}, m_{\text{all}})$ to be the comparable Spearman correlation of the circuit. For each latent, we sum the attribution scores across all functional sequences (denoted by the entry `DMS_score_bin = 1` per mutant) to get a ranking of the top latents, and then proceed with our iterative greedy search method as previously described.

**Plotting.** To create Fig. 1d, Fig. 5b, and Fig. 7, we only include the results where the performance of ESM2 obtains a Spearman correlation of above or equal to 0.01, for similar reasons as in the protein family case. These circuits are used to plot the distribution of latents in Fig. 6b. We leave all function prediction circuits on our HuggingFace.

### D.3. Implementation of Circuit Steering

**Data and Circuits.** We conduct our steering experiments on a subset of DMS assays from Table 2: SPG1_STRSG_Olson_2014, HIS7_YEAST_Pokusaeva_2019, GRB2_HUMAN_Faure_2021, GFP_AEQVI_Sarkisyan_2016, CAPSD_AAV2S_Sinai_2021, RASK_HUMAN_Weng_2022_abundance, and A4_HUMAN_Seuma_2022. We picked these DMS assays due to having more than 10,000 multiple mutants present, which provides more training data for our CNN evaluation model to better learn the fitness landscape. For each of the replacement models, we steer the circuits attained from the multiple mutation groupings.

**CNN Evaluation Model Training.** To create a CNN evaluation model to approximate the fitness landscape, we copy the CNN architecture, hyperparameters, and training strategy from our function prediction circuit CNN probe training methodology. We randomly split the entire DMS assay into a training (90%) and test (10%) set.

**Attribution-Weighted Sampling.** Similar to the function circuit discovery procedure, we rank all latents by descending attribution score. To select a diverse set of influential features, we convert the attribution scores of all circuit latents into a probability distribution. We then sample $C$ many latents without replacement from this distribution for five trials per cross-validation fold. In our experiments, we steer across $C \in [4, 8, 16]$ many latents.

**Steering.** For each DMS assay, we first feed in the wildtype sequence into ESM2. At each token index, we find the feature activations $\mathbf{a}^{\ell}$. If $a_i^{\ell}$ is in the set of latents to be steered, we clamp $a_i^{\ell}$ to $\alpha$ times its maximum activation value across all tokens and all $i \in [d_{\text{latent}}]$.

We then decode the latents using the steered version of the activations $\mathbf{a}^{\ell}$ to obtain the steered reconstruction. To ensure that the model output is affected only by the steering vector and not by the lossy reconstruction error of the replacement model, we apply an error correction term. We first pass the unsteered wildtype sequence through the replacement model to generate a reconstruction, and calculate the difference between the reconstruction and the true ESM2 output. This residual error is then added to the steered output. From here, we continue the forward pass as specified by each replacement model to output the logits.

To determine which amino acids to mutate, we compare the logits of the steered sequence to the logits of the wildtype sequence. Following (Tsui et al., 2025b), at each amino acid position, we compute the cosine similarity between the respective logit vectors. We mutate a position if the cosine similarity is below 0.98, which ensures we only mutate amino acids where ESM2 has made a meaningful change. Additionally, to ensure that our CNN evaluation model is a good proxy for fitness, we constrain our search space to designing protein variants with a maximum of five mutations away. In cases where a steered sequence suggests more than five mutations, we select five mutations based on the largest logit values and set the rest of the sequence to the wildtype.

**Experimental Setup.** For each $C$, we steer our latents over values of $\alpha$ from 0.1 to 5 over 25 evenly spaced steps. For each multiplier, we use our CNN probes from circuit discovery to score each mutated sequence. We then select the top 50 mutated sequences with the highest predicted fitness and score them using our CNN evaluation model. We perform this over all five cross-validation folds and over five trials per cross-validation fold.

To create the random baseline, for each of the steered sequences, we construct a random sequence that has the same number of mutations as the steered sequence, with the positions and mutations chosen uniformly at random. We then randomly select 50 of these sequences and score them using the CNN evaluation model.

### D.4. Implementation of CAA Steering

We add Contrastive Activation Addition (CAA) as a baseline to our steering experiments due to its ability to steer at all layers (as in the case with ProtoMech and PLT). We adapt the CAA (Rimsky et al., 2024) steering method from (Huang

et al., 2025), but modify the algorithm slightly to generate mutations locally around a wildtype sequence. Starting with a given DMS dataset, we divide the dataset into positive and negative sequences based on a threshold. We then run 10 trials, where for each trial we sample 10% of the positive and negative sequences randomly to create positive set $\mathcal{P}$ and negative set $\mathcal{N}$. Following the notation and formulation in (Huang et al., 2025), we generate the steering vector $\mathbf{v}_\ell$ as:

$$\mathbf{v}_\ell = \frac{1}{|\mathcal{P}|} \sum_{x_p \in \mathcal{P}} \mathbf{h}_\ell^{\mathrm{avg}}(x_p) - \frac{1}{|\mathcal{N}|} \sum_{x_n \in \mathcal{N}} \mathbf{h}_\ell^{\mathrm{avg}}(x_n), \tag{10}$$

where $\mathbf{h}_\ell$ represents the mean-pooled ESM activation at layer $\ell$ ($\mathbf{x}^\ell + \mathbf{y}^\ell$) and $x_p \in \mathcal{P}$ and $x_n \in \mathcal{N}$ represent a sequence from the positive and negative set, respectively. We then add the steering vector as:

$$\tilde{\mathbf{h}}_\ell = \mathbf{h}_\ell + \alpha \mathbf{v}_\ell \tag{11}$$

and re-normalize $\tilde{\mathbf{h}}_\ell$ to have the same norm as $\mathbf{h}_\ell$. The addition in Equation (11) is done at each token and each layer. We do this at every layer and decode to the logits. To choose mutations by changing to the amino acid token at each position to the maximum logit probability. In cases where the mutated sequence would be over 5 mutations, we choose the 5 mutations with the highest logit probabilities. We note that we did not using the cosine similarity trick as in steering, as we experienced difficulties generating diverse sequences (see below) and wanted to maximize the amount of sequences generated. We run CAA twice with ten trials each. We first set the positive set to be the set of sequences with a score in the top or bottom 10%, and we then set the the positive sequences as functioning sequences (indicated in ProteinGym by a `DMS_score_bin` of 1) and the negative sequences as nonfunctioning sequences (indicated in ProteinGym by a `DMS_score_bin` of 0). We steer over values of $\alpha$ from 0.1 to 5 over 25 evenly spaced steps.

When running CAA, we observe that a weakness of this method is a lack of diverse sequences generated. We notice that despite steering over multiple trials, CAA fails to produce many unique sequences for some DMS datasets (and in some cases produces only one unique sequence), unlike other methods. Therefore, when reporting our steering results, in cases where the top 10% and 20% score happens to only contain a single sequence (which happens when the rest of the sequences' evaluation scores are significantly worse than the max sequence score), we take the top 10% and 20% sequences, ranked by their score.

## E. Additional Experimental Results

In this section, we detail our experimental results that did not fit the main text. When the ESM2 model is not specified, we default to reporting on ESM2-8M. We leave the majority of our results on ESM2-35M in Appendix G.

### E.1. Circuit Discovery

As noted in Appendix C.2, in the main text (Fig. 1c), we compare the performance of ProtoMech and PLT using the *sequential* replacement model on circuit discovery tasks to ensure a fair comparison of the representational power of both architectures. In this section, we expand our evaluation to include the full spectrum of replacement models across both protein family classification and function prediction tasks (Fig. 5).

We observe that the ProtoMech direct replacement model achieves the highest performance among all other replacement models. We attribute this to the direct replacement model's ability to bypass layer-wise sequential operations, avoiding compounding reconstruction errors. Furthermore, when controlling for the replacement model, ProtoMech consistently outperforms PLT. Lastly, we observe a large gap between the full and sequential replacement models. This phenomenon is consistent with findings in (Ameisen et al., 2025) and, as an aside, we note that developing robust full replacement models remains an active area of research.

Fig. 6 illustrates the average number of latents per layer used in circuit discovery across all replacement models. In ProtoMech, we observe that the latent usage of the direct model is the most sparse, followed by sequential and full replacement. This is unsurprising, as the direct model, is optimized solely to reconstruct $\mathbf{y}^L$ given Equation (2). On the other hand, the full replacement model must retain more latents to support a complete forward pass, requiring information necessary to correctly drive downstream attention mechanisms and maintain signal propagation.

Additionally, we observe a similar phenomenon to (Adams et al., 2025), where the majority of latents being utilized are concentrated in the final layers. We attribute this to the information captured at each layer. In the last layer, we observe that

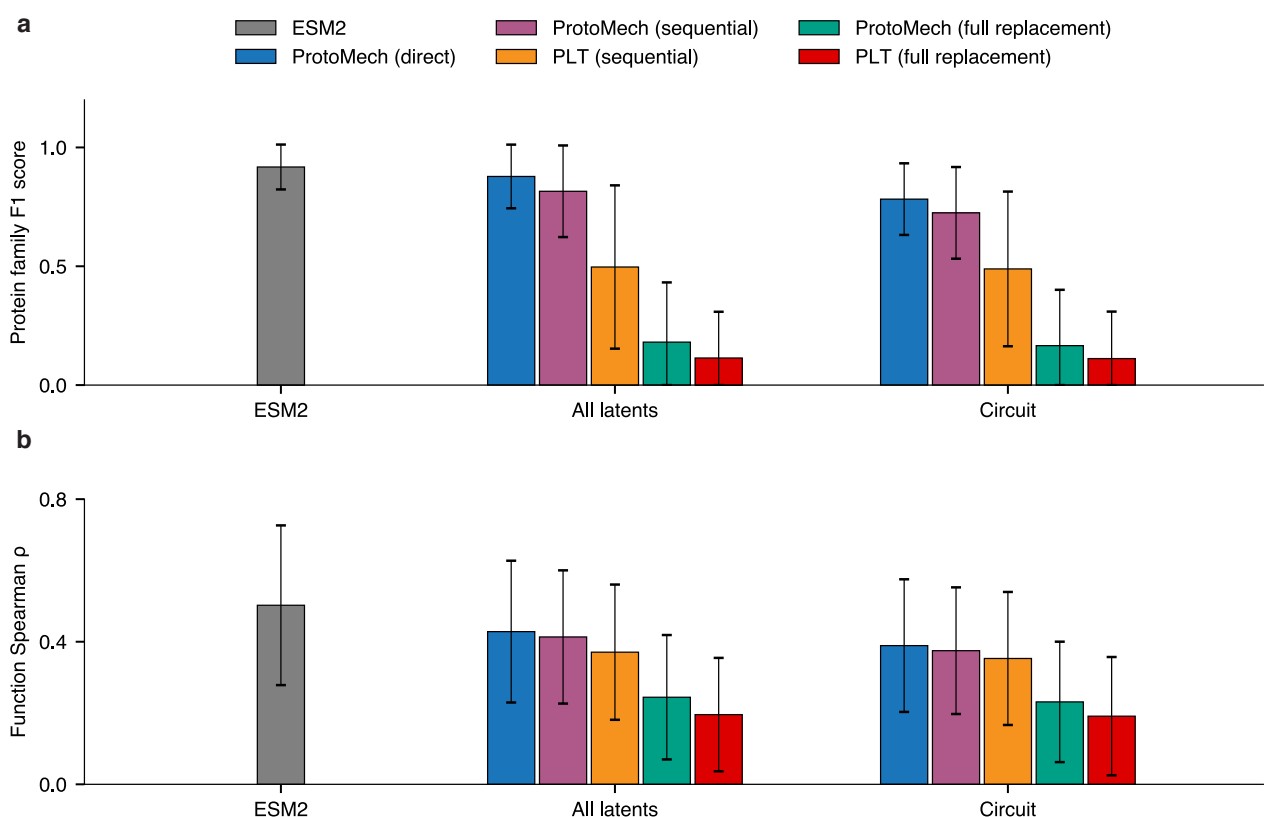

*Figure 5.* Performance of all replacement models on **a**, protein family classification and **b**, function prediction tasks.

latents typically capture granular, amino acid-level biochemical patterns, necessitating a high volume of specific features. In contrast, the middle layers appear to encode higher-level structural motifs, which can be represented more efficiently with a sparser set of latents.

Finally, Fig. 7 details the performance breakdown of our function prediction results. The relative performance of our replacement models remains consistent with Fig. 5.

From here, we perform a preliminary analysis to determine whether the discovered circuits are learning global functional rules. We analyze the GFP_AEQVI_Sarkisyan (Sarkisyan et al., 2016) DMS assay, which contains variants up to 10 mutations away. We stratified the Spearman correlation by mutation depth between ESM2-8M and 35M, ProtoMech with all latents, and the circuit discovered in the first random seed (Tables 3 and 4).

*Table 3.* Comparison of ProtoMech performance on ESM2-8M on the GFP_AEQVI_Sarkisyan DMS assay, stratified against mutation depth.

| Mutation Depth | ESM2-8M | ProtoMech (all latents) | ProtoMech (circuit) |
|---|---|---|---|
| 1 | 0.24 | 0.16 | 0.18 |
| 2 | 0.30 | 0.20 | 0.17 |
| 3 | 0.37 | 0.25 | 0.23 |
| 4 | 0.46 | 0.34 | 0.32 |
| 5+ | 0.35 | 0.27 | 0.26 |

We observe that the circuits discovered by ProtoMech are able to maintain their predictive power across all mutation depths. In particular, at 5+ mutations, ProtoMech circuits recover up to 74% and 81% of the predictive power of ESM2-8M and ESM2-35M, respectively, suggesting that ProtoMech is capturing computational mechanisms representative of global

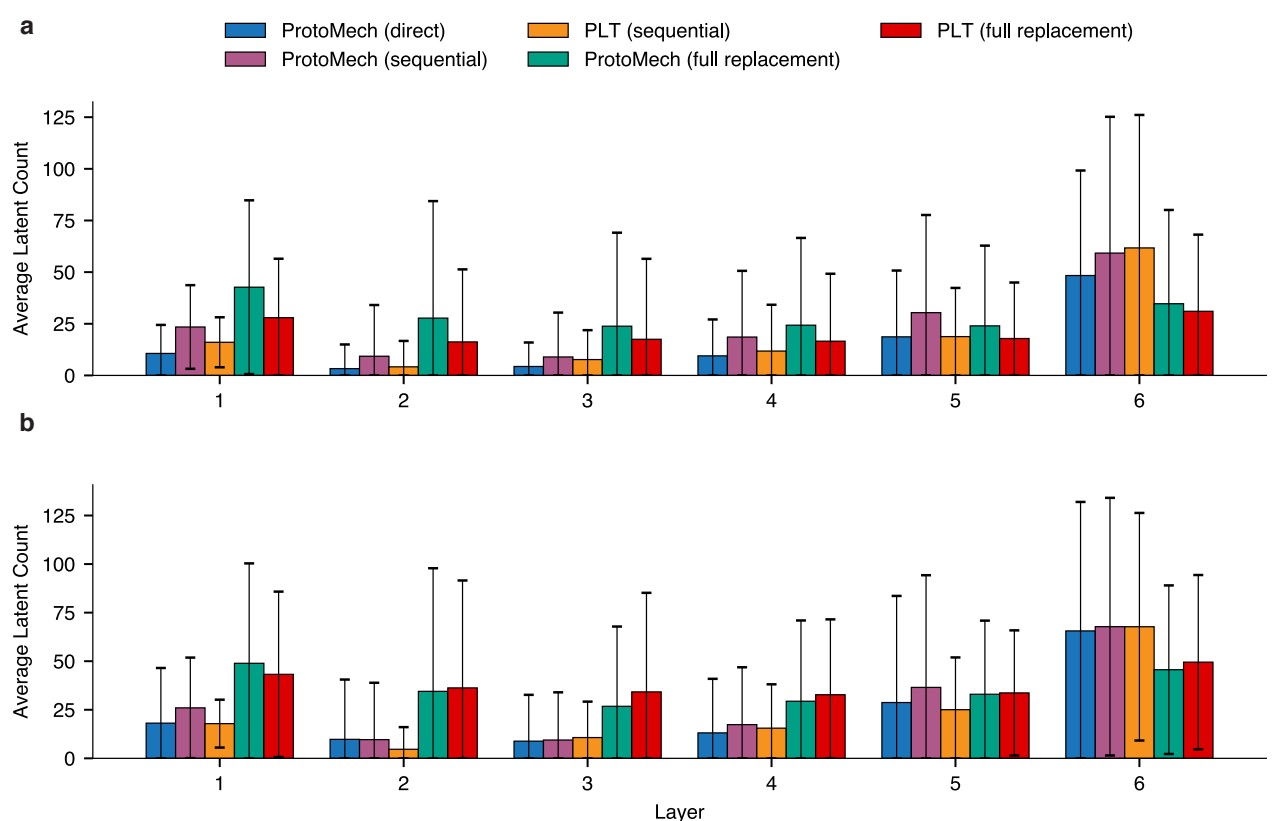

*Figure 6.* Average number of latents in circuits in all replacement models on **a**, protein family classification and **b**, function prediction tasks.

functional rules.

We additionally observe that circuits discovered are stable across various inputs and training folds in family and function circuits. In family circuits, randomly sampling five family sequences from the five largest families, we have an $81\% \pm 21\%$ and $79\% \pm 15\%$ overlap in nodes in ESM2-8M and ESM2-35M, respectively. Similarly, in function circuits, across all 5 folds and all DMS assays in ProteinGym and using the smallest circuit as reference, we have a $96\% \pm 4\%$ and $97\% \pm 4\%$ overlap in nodes in ESM2-8M and ESM2-35M, respectively.

### E.2. Denoising ESM2 in Poor Performance Regimes

We observe that ProtoMech outperforms the original model when ESM2 performs poorly. Fig. 8 details our performance results when ESM2 achieves an F1 score of less than 0.5. Notably, ProtoMech tends to *outperform* the original model on average. Using all latents in the sequential replacement model, ProtoMech attains an F1 of $0.40 \pm 0.18$ compared to ESM2's original performance of $0.39 \pm 0.04$. In circuit discovery, ProtoMech attains an F1 of $0.43 \pm 0.27$. Additionally, in function prediction, while ESM2 attains a higher average performance when its Spearman correlation is below 0.2, we find that 37% of circuits are outperforming it. We attribute this gain to ProtoMech effectively denoising ESM2's computation in regimes where the original model performs poorly.

### E.3. Steering

To determine the optimal intervention strategy for circuit steering, we conducted an ablation study comparing different replacement model architectures. To establish the PLT baseline, we first compared the performance of PLT circuits using the sequential and full replacement models (Table 5). We observe that the sequential replacement model yields superior fitness outcomes, outperforming the full replacement model in 57% of the evaluated cases. Hence, we utilize the sequential replacement model as our PLT baseline in Table 1.

*Table 4.* Comparison of ProtoMech performance on ESM2-35M on the GFP_AEQVI_Sarkisyan DMS assay, stratified against mutation depth.

| Mutation Depth | ESM2-35M | ProtoMech (all latents) | ProtoMech (circuit) |
|---|---|---|---|
| 1 | 0.21 | 0.18 | 0.15 |
| 2 | 0.29 | 0.24 | 0.16 |
| 3 | 0.37 | 0.32 | 0.23 |
| 4 | 0.40 | 0.34 | 0.25 |
| 5+ | 0.31 | 0.29 | 0.25 |

*Table 5.* Steering results using PLT (sequential) and PLT (full replacement). All variants were constrained to a maximum of five mutations away from the wildtype.

| Method | DMS | Mean score ↑ | Max score ↑ | Top 10% score ↑ | Top 20% score ↑ |
|---|---|---|---|---|---|
| **PLT (sequential)** | SPG1_STRSG_Olson_2014 | **1.97 ± 0.48** | **3.18** | **2.88 ± 0.20** | **2.67 ± 0.25** |
| | HIS7_YEAST_Pokusaeva_2019 | **1.27 ± 0.06** | 1.41 | 1.38 ± 0.02 | **1.36 ± 0.03** |
| | GRB2_HUMAN_Faure_2021 | **-0.18 ± 0.14** | **0.15** | **0.12 ± 0.05** | **0.03 ± 0.09** |
| | GFP_AEQVI_Sarkisyan_2016 | 4.40 ± 0.22 | 5.15 | 4.92 ± 0.19 | 4.76 ± 0.21 |
| | CAPSD_AAV2S_Sinai_2021 | **0.81 ± 0.53** | **2.05** | **1.87 ± 0.11** | **1.67 ± 0.22** |
| | RASK_HUMAN_Weng_2022_abundance | -0.19 ± 0.08 | **0.06** | **-0.02 ± 0.06** | **-0.07 ± 0.07** |
| | A4_HUMAN_Seuma_2022 | -0.05 ± 0.38 | 1.03 | 0.67 ± 0.24 | 0.52 ± 0.24 |
| **PLT (full replacement)** | SPG1_STRSG_Olson_2014 | 1.40 ± 0.61 | **3.18** | 2.57 ± 0.33 | 2.30 ± 0.37 |
| | HIS7_YEAST_Pokusaeva_2019 | 1.16 ± 0.12 | **1.52** | **1.40 ± 0.07** | 1.34 ± 0.08 |
| | GRB2_HUMAN_Faure_2021 | **-0.18 ± 0.13** | 0.12 | 0.09 ± 0.02 | **0.03 ± 0.08** |
| | GFP_AEQVI_Sarkisyan_2016 | **4.52 ± 0.25** | **5.16** | **5.00 ± 0.11** | **4.89 ± 0.14** |
| | CAPSD_AAV2S_Sinai_2021 | 0.77 ± 0.41 | 1.82 | 1.68 ± 0.08 | 1.47 ± 0.23 |
| | RASK_HUMAN_Weng_2022_abundance | **-0.17 ± 0.08** | -0.01 | -0.05 ± 0.02 | **-0.07 ± 0.03** |
| | A4_HUMAN_Seuma_2022 | **0.40 ± 0.37** | **1.51** | **1.12 ± 0.22** | **0.94 ± 0.24** |

With the PLT baseline established, we evaluated the direct, sequential, and full replacement models in ProtoMech. We benchmarked each configuration against the PLT sequential baseline, CAA, and selecting random mutations. We additionally note that, as stated in the Appendix D.3, the number of mutations for the random baseline is based on the number of mutations made in the respective ProtoMech replacement models. As such, the random baseline is expected to fluctuate depending on the ProtoMech replacement model being evaluated. Our results indicate that ProtoMech consistently outperforms the PLT baseline across all configurations, although the margin of improvement varies. Specifically, the ProtoMech direct, sequential, and full replacement models outperformed the PLT baseline in 71% (Table 1), 61% (Table 6), and 57% (Table 7) of cases, respectively, suggesting that the direct model is most suitable for steering ProtoMech.

In addition to PLT steering and CAA, both baselines which utilize the internal representations of ESM2, we also perform an ablation study benchmarking against selecting mutations via ESM2's logits (Table 8). We observe that ProtoMech consistently outperforms suggested ESM2 logits, further validating our discovered mechanisms.

Additionally, we assess the performance of ProtoMech steering comparing against PEX (Ren et al., 2022) and GGS (Kirjner et al., 2024), two black-box optimization methods, on the SPG1_STRSG_Olson_2014 and HIS7_YEAST_Pokusaeva_2019 DMS assays (Table 9). We ran PEX and GGS for 10 rounds under default settings using the respective circuit discovery CNN models as the black-box fitness function and then evaluated the fitness of the top 50 sequences from each method using our evaluation CNN model. Interestingly, sequences generated via ProtoMech-based circuit steering outperform those produced by both PEX and GGS across all reported metrics. This is notable given that ProtoMech is not designed as an optimizer, yet it yields superior functional sequences. These results demonstrate that the circuits identified by ProtoMech are not only interpretable but also capture actionable, functionally meaningful mechanisms.

We use one of the high-fitness proteins designed by the direct replacement model over GB1 to display in Fig. 2. The structure of the variant was generated via AlphaFold3 (Abramson et al., 2024).

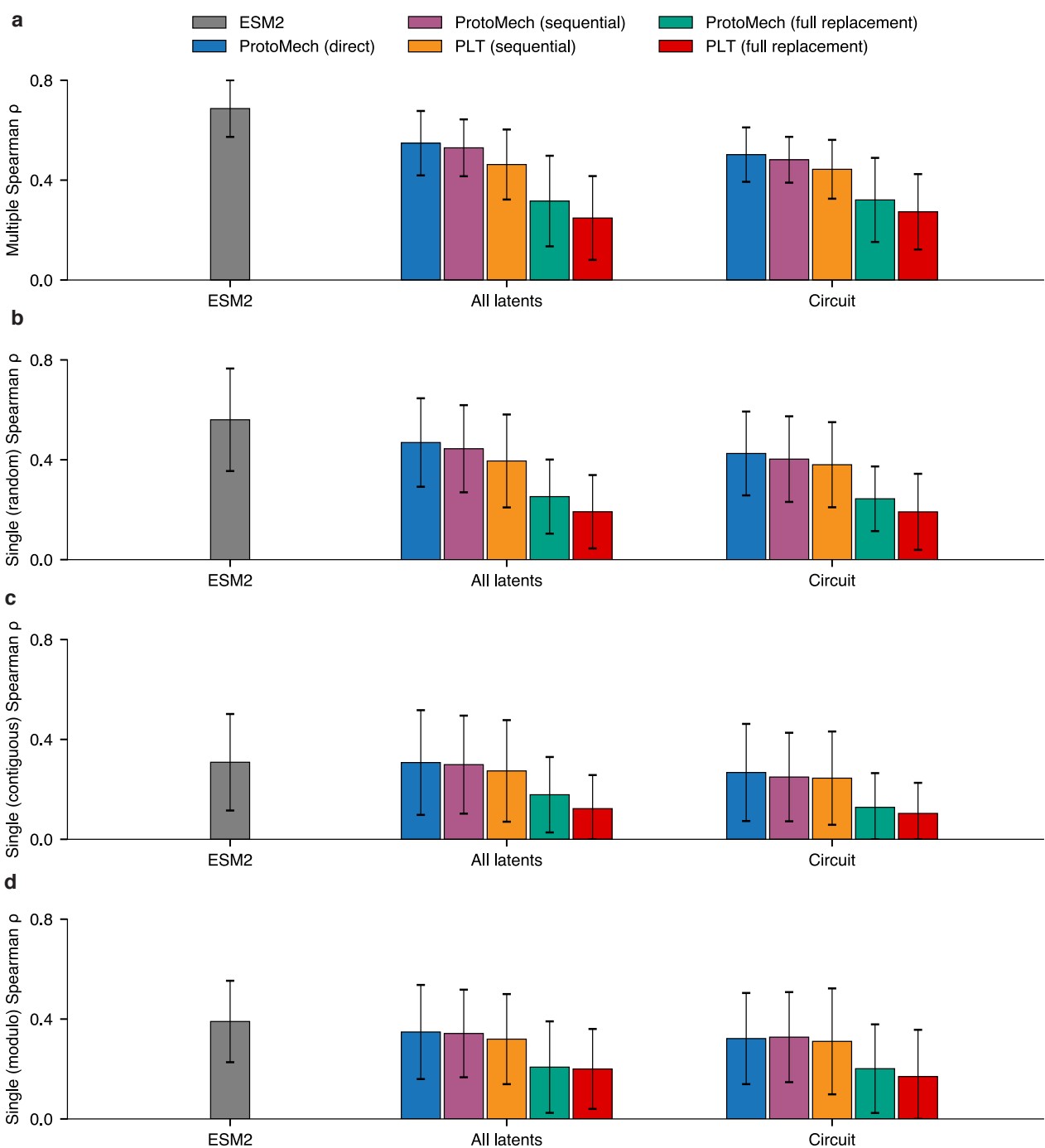

*Figure 7.* Performance of all replacement models on **a**, mutliple, **b**, single (random), **c**, single (contiguous), and **d**, single (modulo), function prediction tasks.

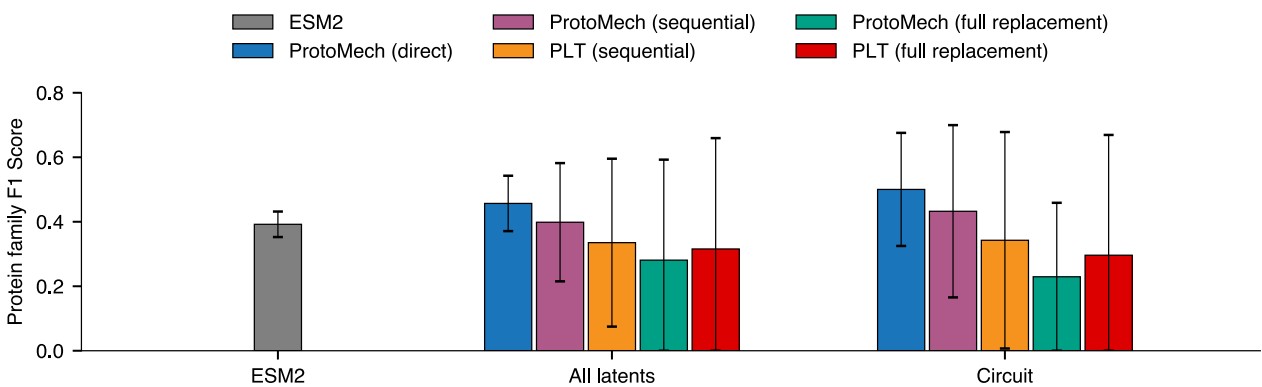

*Figure 8.* Performance of all replacement models on protein family classification where ESM2 achieved an F1 score of less than 0.5.

*Table 6.* Steering results using ProtoMech (sequential), PLT (sequential), CAA, and selecting random mutations on seven DMS assays. All variants were constrained to a maximum of five mutations away from the wildtype.

| Method | DMS | Mean score ↑ | Max score ↑ | Top 10% score ↑ | Top 20% score ↑ |
|---|---|---|---|---|---|
| **ProtoMech (sequential)** | SPG1_STRSG_Olson_2014 | $1.93 \pm 0.55$ | **3.24** | **2.98 ± 0.22** | **2.78 ± 0.25** |
| | HIS7_YEAST_Pokusaeva_2019 | **1.28 ± 0.09** | **1.50** | **1.45 ± 0.03** | **1.41 ± 0.04** |
| | GRB2_HUMAN_Faure_2021 | **-0.17 ± 0.11** | **0.15** | $0.06 \pm 0.06$ | $0.01 \pm 0.07$ |
| | GFP_AEQVI_Sarkisyan_2016 | $3.85 \pm 0.21$ | 4.38 | $4.31 \pm 0.06$ | $4.20 \pm 0.12$ |
| | CAPSD_AAV2S_Sinai_2021 | $0.37 \pm 0.33$ | 1.35 | $1.05 \pm 0.17$ | $0.88 \pm 0.22$ |
| | RASK_HUMAN_Weng_2022_abundance | $-0.04 \pm 0.09$ | **0.20** | **0.13 ± 0.04** | **0.10 ± 0.04** |
| | A4_HUMAN_Seuma_2022 | **0.33 ± 0.42** | **1.53** | **1.30 ± 0.26** | **1.00 ± 0.36** |
| **PLT (sequential)** | SPG1_STRSG_Olson_2014 | **1.97 ± 0.48** | 3.18 | $2.88 \pm 0.20$ | $2.67 \pm 0.25$ |
| | HIS7_YEAST_Pokusaeva_2019 | $1.27 \pm 0.06$ | 1.41 | $1.38 \pm 0.02$ | $1.36 \pm 0.03$ |
| | GRB2_HUMAN_Faure_2021 | $-0.18 \pm 0.14$ | **0.15** | **0.12 ± 0.05** | **0.03 ± 0.09** |
| | GFP_AEQVI_Sarkisyan_2016 | **4.40 ± 0.22** | **5.15** | **4.92 ± 0.19** | **4.76 ± 0.21** |
| | CAPSD_AAV2S_Sinai_2021 | **0.81 ± 0.53** | **2.05** | **1.87 ± 0.11** | **1.67 ± 0.22** |
| | RASK_HUMAN_Weng_2022_abundance | $-0.19 \pm 0.08$ | 0.06 | $-0.02 \pm 0.06$ | $-0.07 \pm 0.07$ |
| | A4_HUMAN_Seuma_2022 | $-0.05 \pm 0.38$ | 1.03 | $0.67 \pm 0.24$ | $0.52 \pm 0.24$ |
| **CAA** | SPG1_STRSG_Olson_2014 | $0.70 \pm 0.00$ | 0.70 | $0.70 \pm 0.00$ | $0.70 \pm 0.00$ |
| | HIS7_YEAST_Pokusaeva_2019 | $0.52 \pm 0.14$ | 0.77 | $0.77 \pm 0.00$ | $0.71 \pm 0.06$ |
| | GRB2_HUMAN_Faure_2021 | $-0.40 \pm 0.11$ | -0.20 | $-0.21 \pm 0.02$ | $-0.23 \pm 0.03$ |
| | GFP_AEQVI_Sarkisyan_2016 | $2.93 \pm 0.23$ | 3.16 | $3.16 \pm 0.00$ | $3.16 \pm 0.00$ |
| | CAPSD_AAV2S_Sinai_2021 | $-0.26 \pm 0.55$ | 0.45 | $0.45 \pm 0.00$ | $0.45 \pm 0.00$ |
| | RASK_HUMAN_Weng_2022_abundance | $-0.35 \pm 0.07$ | -0.28 | $-0.28 \pm 0.00$ | $-0.29 \pm 0.01$ |
| | A4_HUMAN_Seuma_2022 | $-1.90 \pm 0.03$ | -1.87 | $-1.87 \pm 0.00$ | $-1.87 \pm 0.00$ |
| **Random** | SPG1_STRSG_Olson_2014 | $-2.57 \pm 0.82$ | -1.15 | $-1.36 \pm 0.16$ | $-1.54 \pm 0.21$ |
| | HIS7_YEAST_Pokusaeva_2019 | $0.56 \pm 0.24$ | 1.08 | $0.98 \pm 0.06$ | $0.91 \pm 0.08$ |
| | GRB2_HUMAN_Faure_2021 | $-0.84 \pm 0.23$ | -0.45 | $-0.49 \pm 0.03$ | $-0.53 \pm 0.05$ |
| | GFP_AEQVI_Sarkisyan_2016 | $2.76 \pm 0.46$ | 3.72 | $3.51 \pm 0.18$ | $3.35 \pm 0.21$ |
| | CAPSD_AAV2S_Sinai_2021 | $-1.02 \pm 0.53$ | 0.27 | $0.10 \pm 0.14$ | $-0.21 \pm 0.33$ |
| | RASK_HUMAN_Weng_2022_abundance | $-0.69 \pm 0.25$ | -0.23 | $-0.31 \pm 0.05$ | $-0.36 \pm 0.06$ |
| | A4_HUMAN_Seuma_2022 | $-1.41 \pm 0.47$ | -0.43 | $-0.68 \pm 0.21$ | $-0.87 \pm 0.25$ |

*Table 7.* Steering results using ProtoMech (full replacement), PLT (sequential), CAA, and selecting random mutations on seven DMS assays. All variants were constrained to a maximum of five mutations away from the wildtype.

| Method | DMS | Mean score ↑ | Max score ↑ | Top 10% score ↑ | Top 20% score ↑ |
|---|---|---|---|---|---|
| **ProtoMech (full replacement)** | SPG1_STRSG_Olson_2014 | 1.39 ± 0.68 | **3.18** | 2.67 ± 0.30 | 2.38 ± 0.36 |
| | HIS7_YEAST_Pokusaeva_2019 | 1.14 ± 0.08 | 1.33 | 1.28 ± 0.04 | 1.24 ± 0.05 |
| | GRB2_HUMAN_Faure_2021 | **-0.15 ± 0.13** | 0.13 | 0.08 ± 0.04 | **0.04 ± 0.05** |
| | GFP_AEQVI_Sarkisyan_2016 | **4.64 ± 0.45** | **5.70** | **5.47 ± 0.12** | **5.34 ± 0.17** |
| | CAPSD_AAV2S_Sinai_2021 | **1.07 ± 0.57** | 3.28 | 2.40 ± 0.47 | 1.94 ± 0.58 |
| | RASK_HUMAN_Weng_2022_abundance | **-0.13 ± 0.09** | 0.05 | 0.02 ± 0.01 | 0.00 ± 0.02 |
| | A4_HUMAN_Seuma_2022 | -0.13 ± 0.35 | **1.18** | 0.67 ± 0.30 | 0.41 ± 0.34 |
| **PLT (sequential)** | SPG1_STRSG_Olson_2014 | **1.97 ± 0.48** | **3.18** | **2.88 ± 0.20** | **2.67 ± 0.25** |
| | HIS7_YEAST_Pokusaeva_2019 | **1.27 ± 0.06** | **1.41** | **1.38 ± 0.02** | **1.36 ± 0.03** |
| | GRB2_HUMAN_Faure_2021 | -0.18 ± 0.14 | **0.15** | **0.12 ± 0.05** | 0.03 ± 0.09 |
| | GFP_AEQVI_Sarkisyan_2016 | 4.40 ± 0.22 | 5.15 | 4.92 ± 0.19 | 4.76 ± 0.21 |
| | CAPSD_AAV2S_Sinai_2021 | 0.81 ± 0.53 | 2.05 | 1.87 ± 0.11 | 1.67 ± 0.22 |
| | RASK_HUMAN_Weng_2022_abundance | -0.19 ± 0.08 | **0.06** | -0.02 ± 0.06 | -0.07 ± 0.07 |
| | A4_HUMAN_Seuma_2022 | **-0.05 ± 0.38** | 1.03 | **0.67 ± 0.24** | **0.52 ± 0.24** |
| **CAA** | SPG1_STRSG_Olson_2014 | 0.70 ± 0.00 | 0.70 | 0.70 ± 0.00 | 0.70 ± 0.00 |
| | HIS7_YEAST_Pokusaeva_2019 | 0.52 ± 0.14 | 0.77 | 0.77 ± 0.00 | 0.71 ± 0.06 |
| | GRB2_HUMAN_Faure_2021 | -0.40 ± 0.11 | -0.20 | -0.21 ± 0.02 | -0.23 ± 0.03 |
| | GFP_AEQVI_Sarkisyan_2016 | 2.93 ± 0.23 | 3.16 | 3.16 ± 0.00 | 3.16 ± 0.00 |
| | CAPSD_AAV2S_Sinai_2021 | -0.26 ± 0.55 | 0.45 | 0.45 ± 0.00 | 0.45 ± 0.00 |
| | RASK_HUMAN_Weng_2022_abundance | -0.35 ± 0.07 | -0.28 | -0.28 ± 0.00 | -0.29 ± 0.01 |
| | A4_HUMAN_Seuma_2022 | -1.90 ± 0.03 | -1.87 | -1.87 ± 0.00 | -1.87 ± 0.00 |
| **Random** | SPG1_STRSG_Olson_2014 | -2.65 ± 1.04 | -0.52 | -0.85 ± 0.30 | -1.20 ± 0.42 |
| | HIS7_YEAST_Pokusaeva_2019 | 0.44 ± 0.33 | 0.86 | 0.83 ± 0.01 | 0.79 ± 0.05 |
| | GRB2_HUMAN_Faure_2021 | -0.82 ± 0.30 | -0.33 | -0.40 ± 0.06 | -0.45 ± 0.07 |
| | GFP_AEQVI_Sarkisyan_2016 | 2.79 ± 0.47 | 3.80 | 3.48 ± 0.23 | 3.35 ± 0.21 |
| | CAPSD_AAV2S_Sinai_2021 | -1.04 ± 0.66 | 0.24 | 0.03 ± 0.13 | -0.14 ± 0.19 |
| | RASK_HUMAN_Weng_2022_abundance | -0.67 ± 0.32 | -0.04 | -0.18 ± 0.12 | -0.27 ± 0.13 |
| | A4_HUMAN_Seuma_2022 | -1.40 ± 0.49 | -0.52 | -0.78 ± 0.15 | -0.90 ± 0.16 |

*Table 8.* Steering results using ProtoMech (direct) compared to ESM2 logits on seven DMS assays. All variants were constrained to a maximum of five mutations away from the wildtype.

| Method | DMS | Mean score ↑ | Max score ↑ | Top 10% score ↑ | Top 20% score ↑ |
|---|---|---|---|---|---|
| **ProtoMech (direct)** | SPG1_STRSG_Olson_2014 | **1.67 ± 0.67** | **3.24** | **3.00 ± 0.21** | **2.70 ± 0.34** |
| | HIS7_YEAST_Pokusaeva_2019 | **1.28 ± 0.09** | **1.42** | **1.41 ± 0.01** | **1.40 ± 0.02** |
| | GRB2_HUMAN_Faure_2021 | **-0.15 ± 0.07** | 0.05 | **-0.01 ± 0.04** | **-0.05 ± 0.04** |
| | GFP_AEQVI_Sarkisyan_2016 | **4.17 ± 0.23** | **4.63** | **4.56 ± 0.05** | **4.50 ± 0.08** |
| | CAPSD_AAV2S_Sinai_2021 | **1.68 ± 1.08** | **4.54** | **3.77 ± 0.43** | **3.36 ± 0.53** |
| | RASK_HUMAN_Weng_2022_abundance | **-0.12 ± 0.06** | **0.06** | **0.02 ± 0.03** | **-0.02 ± 0.05** |
| | A4_HUMAN_Seuma_2022 | **0.12 ± 0.33** | **1.07** | **0.76 ± 0.18** | **0.63 ± 0.19** |
| **ESM logits** | SPG1_STRSG_Olson_2014 | -0.72 ± 1.42 | 2.45 | 2.04 ± 0.28 | 1.47 ± 0.64 |
| | HIS7_YEAST_Pokusaeva_2019 | 0.73 ± 0.24 | 1.28 | 1.23 ± 0.06 | 1.08 ± 0.17 |
| | GRB2_HUMAN_Faure_2021 | -0.54 ± 0.25 | 0.12 | -0.06 ± 0.09 | -0.12 ± 0.10 |
| | GFP_AEQVI_Sarkisyan_2016 | 3.02 ± 0.42 | 4.17 | 3.98 ± 0.21 | 3.67 ± 0.35 |
| | CAPSD_AAV2S_Sinai_2021 | -1.06 ± 0.25 | -0.62 | -0.70 ± 0.06 | -0.76 ± 0.07 |
| | RASK_HUMAN_Weng_2022_abundance | -0.39 ± 0.09 | -0.17 | -0.22 ± 0.04 | -0.27 ± 0.06 |
| | A4_HUMAN_Seuma_2022 | -1.61 ± 0.93 | 0.07 | -0.18 ± 0.17 | -0.42 ± 0.29 |

*Table 9.* Steering results comparison between ProtoMech, PEX, and GGS across DMS assays. Bold values indicate the best performance for each specific protein variant.

| Method | DMS | Mean score ↑ | Max score ↑ | Top 10% score ↑ | Top 20% score ↑ |
|---|---|---|---|---|---|
| **ProtoMech (direct)** | SPG1_STRSG_Olson_2014 | **1.67 ± 0.67** | **3.24** | **3.00 ± 0.21** | **2.70 ± 0.34** |
| | HIS7_YEAST_Pokusaeva_2019 | **1.28 ± 0.09** | **1.42** | **1.41 ± 0.01** | **1.40 ± 0.02** |
| **PEX** | SPG1_STRSG_Olson_2014 | 1.50 ± 0.32 | 2.37 | 2.12 ± 0.16 | 1.95 ± 0.22 |
| | HIS7_YEAST_Pokusaeva_2019 | 0.86 ± 0.10 | 1.09 | 1.04 ± 0.03 | 1.01 ± 0.04 |
| **GGS** | SPG1_STRSG_Olson_2014 | 0.65 ± 0.40 | 1.33 | 1.25 ± 0.05 | 1.19 ± 0.08 |
| | HIS7_YEAST_Pokusaeva_2019 | 0.74 ± 0.06 | 0.87 | 0.84 ± 0.02 | 0.82 ± 0.03 |

# F. ProtoMech Visualization

In this section, we detail how we visualize circuits in the paper and the main functions of our visualization tool.

## F.1. Visualizing Circuits

In ProtoMech, we follow the procedure according to (Ameisen et al., 2025) in order to visualize our circuits. Notably, this involves creating a *local replacement model*, which creates a unique replacement model for each sequence.

**Local Replacement.** The local replacement model uses a base sequence $s$ and assumes access to all ground-truth inputs and outputs $\mathbf{x}_s^\ell, \mathbf{y}_s^\ell$ for $s$. Given an input sequence, the update rule becomes:

$$\mathbf{x}^\ell = \hat{\mathbf{x}}_{\text{pre}}^\ell + \sum_{\tilde{h} \in H_\ell} \tilde{h}(\mathbf{x}_{\text{pre}}^\ell), \tag{12}$$

where $\tilde{h}(\mathbf{x}_{\text{pre}}^\ell)$ represents the output of the attention head $h$ with its attention pattern frozen to the value of the attention pattern from the base model. The denominators of the LayerNorm functions are fixed to the standard deviation of $\mathbf{x}_s^\ell$ for each $\ell$. We also add the error between the MLP output $\mathbf{y}^\ell$ and the CLT approximation $\hat{\mathbf{y}}^\ell$ at each layer. By adding in the residual error at every layer, Equation (8) becomes:

$$\mathbf{x}_{\text{pre}}^{\ell+1} = \mathbf{x}^\ell + \hat{\mathbf{y}}^\ell + (\mathbf{y}_s^\ell - \hat{\mathbf{y}}_s^\ell), \tag{13}$$

where $\hat{\mathbf{y}}_s^\ell$ is the CLT reconstruction of $\mathbf{y}_s^\ell$. If the input sequence is $s$ itself, the output is the same as the output of the base model. Some implementation details are adapted from (Krzus, 2025).

**Visualizing Latents.** Using the entire Swiss-Prot dataset, we identify the top-10 maximally activating sequences from Swiss-Prot in order to identify motifs or family-specific patterns that latents consistently light up (available through our ProtoMech visualizer). This complements us analyzing the activations of latents when analyzing the input sequence and allows for a more holistic view of identifying meaningful latents. To create Fig. 3, we feed in the canonical sequences associated with the UniProt IDs P83104 and Q5RKL5 for the protein kinase and NADP+ binding domains, respectively, and create the local replacement model using the circuits discovered from the sequential replacement model. We project the amino acid activations onto their Swiss-Prot structure. To create Fig. 4, we feed in the GB1 domain defined by (Olson et al., 2014) and project the amino acid activations onto the PDB structure `1pga` (Gallagher et al., 1994).

**Edge Weight Computation.** To find the edge weight between a given source and target latent, we use the same method as detailed in Appendix E of (Ameisen et al., 2025) for computing the edge weights (also called virtual weights, which we use interchangeably) for edges between feature latents. Formally, we can write the edge weight between a source latent $s$ and target latent $t$ as

$$A_{s \to t} = a_s w_{s \to t} = a_s \sum_{\ell_s \leq \ell \leq \ell_t} (W_{\text{dec},s}^{\ell_s \to \ell})^T J_{t_s, \ell_s \to t_t, \ell_t}^{\blacktriangledown} W_{\text{enc},t}^{\ell_t},$$

where $J_{t_s, \ell_s \to t_t, \ell_t}^{\blacktriangledown}$ represents the Jacobian between token $s$ at layer $\ell_s$ and token $t$ at layer $\ell_t$, with frozen attention patterns and layernorm denominators. Intuitively, $w_{s \to t}$ computes the gradient between the source activation $a_s$ and the target pre-activation $z_t$, where $\mathbf{z}^{\ell_t} := \mathbf{W}_{\text{enc}}^{\ell_t}(\mathbf{x}^{\ell_t} - \mathbf{b}_{\text{pre}}^{\ell_t}) + \mathbf{b}_{\text{enc}}^{\ell_t}$.

## F.2. Visualization Tool

We provide the ProtoMech visualization tool alongside our paper submission. The platform enables users to interactively explore the circuit structures presented in the paper and analyze new protein sequences. Users can either select from pre-loaded examples or upload custom circuit data by providing the following files:

1. `activation_indices.json`: Contains latents (specified by layer and latent index) that activate at each position in the sequence.

2. `seq.txt`: Contains the full protein sequence.

3. `top_activations.json`: Contains the top-activating sequences for each latent.

4. `virtual_weights.json` (optional): Contains all virtual weights between pairs of latents.

**Main Interface.**    The primary view of ProtoMech displays a grid layout where rows correspond to CLT layers and columns correspond to sequence positions (Figure 9). Each cell contains the latents that activate at that position, with activation intensity encoded by color. When virtual weights are enabled, edges connect latents across layers, with blue indicating positive weights and orange indicating negative weights. The protein sequence is displayed along the bottom axis, allowing users to directly relate latent activations to specific amino acid residues.

**Exploring Latent Rankings.**    Clicking on a layer label (e.g., "Layer 2") opens the Latent Rankings panel, which displays the top-activating latents for that layer ranked by their maximum activation value (Figure 10). Each entry shows the latent index, maximum activation score, the sequence position where maximum activation occurs, and a sequence context window with the activation site highlighted. Users can directly add latents to the canvas or access detailed feature information from this panel.

**Latent Information Panel.**    Clicking on any latent opens a detailed information panel with three tabs (Figures 11–13):

- *Sequences* (Figure 11): Displays the input sequence with a heatmap overlay showing activation intensities across all positions. The current position's amino acid and activation value are shown at the top. Below, the top-activating sequences from Swiss-Prot are listed with their activation scores, allowing users to identify common patterns that drive latent activation.

- *Alignment* (Figure 12): Presents an alignment view where the input sequence is aligned with the top-activating sequences from Swiss-Prot. Activation intensities are overlaid as colored highlights, enabling users to identify conserved motifs and structural features that the latent has learned to recognize.

- *Influences* (Figure 13): Lists all incoming connections (from earlier layers) and outgoing connections (to later layers) for the selected latent. Each connection displays the source/target layer and latent index, the virtual weight magnitude (positive in green, negative in orange), and the number of sequence positions where this connection is active. This view enables systematic exploration of how information flows through the circuit.

**Building and Analyzing Circuits on the Canvas.**    The canvas provides an interactive workspace for constructing and visualizing circuit diagrams (Figure 14). Users can add latents to the canvas by right-clicking on any latent in the grid view or using the "Add to Canvas" button in the Latent Info panel. Once on the canvas, nodes can be freely repositioned and annotated with descriptive labels (e.g., "Glycine-rich loop", "ATP-binding site") to document hypothesized functions. Virtual weight edges are automatically drawn between canvas nodes, with edge color indicating sign (blue for positive, orange for negative) and thickness proportional to magnitude. The "Filter Virtual Weights" slider controls the minimum weight threshold for displayed edges. Canvas layouts can be saved and loaded for reproducibility, and exported as images for publication.

**Interpreting Virtual Weights.**    The edge weight shown between two latents on the canvas represents the average virtual weight across all sequence positions where both latents are active. For example, if latents L1/1983 and L5/1774 co-activate at multiple positions with varying weights, the displayed edge weight is their mean. Users can click the "View" button in the Influences tab to examine position-specific weights in detail.

## G. Scaling ProtoMech

In the main text, we report results on ProtoMech on ESM2-8M. Here, we report results scaling up ProtoMech to ESM2-35M, where $L = 12$ and $d_{\text{model}} = 480$. We set $d_{\text{latent}} = 4800$, which is a $10\times$ expansion factor.

### G.1. Family Circuit Discovery

Figure 15a details the performance of all ESM2-35M replacement models on family circuit discovery. As expected, the performance of the original model increases, with an average F1 score of $0.94 \pm 0.07$. Using the full set of latents, ProtoMech with the sequential replacement model matches the performance of the original model, with similarly an average F1 score of $0.94 \pm 0.07$, suggesting that scaling ProtoMech results in more faithful approximations of the model. This significantly outperforms the PLT baseline, which achieved an average F1 score of $0.69 \pm 0.28$. On circuit discovery, the sequential replacement model achieved an F1 score of $0.80 \pm 0.12$ using $109 \pm 144$ nodes ($2.2\%$ of the total latent space), compared to the PLT baseline, which achieved an F1 score of $0.64 \pm 0.24$ using $256 \pm 238$ nodes.

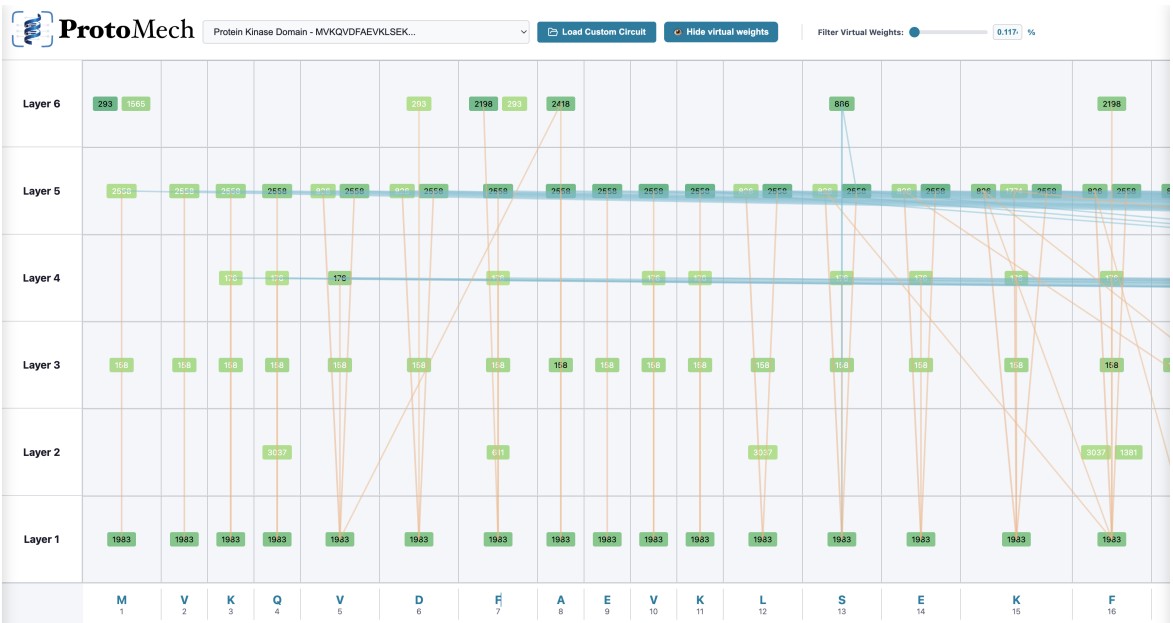

*Figure 9.* **ProtoMech main interface.** The grid view displays latent activations organized by layer (rows) and sequence position (columns). Each colored box represents an active latent, with the latent index shown inside. When virtual weights are enabled, edges connect latents across layers: blue edges indicate positive weights (activation) and orange edges indicate negative weights (inhibition). The protein sequence is displayed along the bottom axis. The toolbar provides access to example circuits, custom data loading, and virtual weight filtering controls.

### G.2. Function Circuit Discovery

Figure 15b details the performance of all ESM2-35M replacement models on function circuit discovery. Similarly, the performance of the original model increases compared to ESM2-8M, with an average Spearman correlation of $0.53 \pm 0.23$. Using the full set of latents, ProtoMech with the sequential replacement model achieves an average Spearman correlation of $0.46 \pm 0.20$, outperforming the PLT baseline ($0.38 \pm 0.19$). On circuit discovery, the sequential replacement model achieved a Spearman correlation of $0.38 \pm 0.19$ using $202 \pm 222$ nodes ($4.2\%$ of the total latent space), compared to the PLT baseline, which achieved a Spearman correlation of $0.33 \pm 0.18$ using $316 \pm 277$ nodes.

We additionally observe a similar denoising effect mentioned in the Discussion. In family circuits where the performance of ESM2-8M is low ($0.49 \pm 0.03$), ProtoMech circuits achieve an F1 of $0.68 \pm 0.23$. Additionally, 40% of function circuits achieve a higher performance than ESM2-35M when the performance is low (under a Spearman of 0.2). These results further reinforce the significance of ProtoMech and its ability to offer utility beyond just interpretability.

### G.3. Windowed CLTs

As mentioned in the Discussion, the computational scalablity of CLTs remains difficult, due to scale with $\mathcal{O}(L^2)$ many decoder matrices. To combat this, we propose "windowed CLTs", which aim to approximate the cross-layer connectivity power of CLTs in localized windows. Here, the encoder mapping via Equation (1) remains identical to the standard CLT architecture. However, the decoder is modified to constrain the reconstruction of layer $\ell$ to a localized span of $w$ preceding layers, where $w$ is a user-defined hyperparameter:

$$\hat{\mathbf{y}}^{\ell} = \sum_{\ell'=\max(1,\ell-w+1)}^{\ell} \mathbf{W}_{\text{dec}}^{\ell' \to \ell} \mathbf{a}^{\ell'} + \mathbf{b}_{\text{pre}}^{\ell}. \tag{14}$$

For example, in ESM2-35M, which has 12 layers, we constrain the reconstruction of $\mathbf{y}^{12}$ to a localized window of the $w = 4$ immediate preceding layers, rather than use every single layer. By doing this, we obtain the following estimated training times for larger ESM2 models in Table 10, extrapolating from our own data. These times are estimated on a single NVIDIA RTX A6000 GPU.

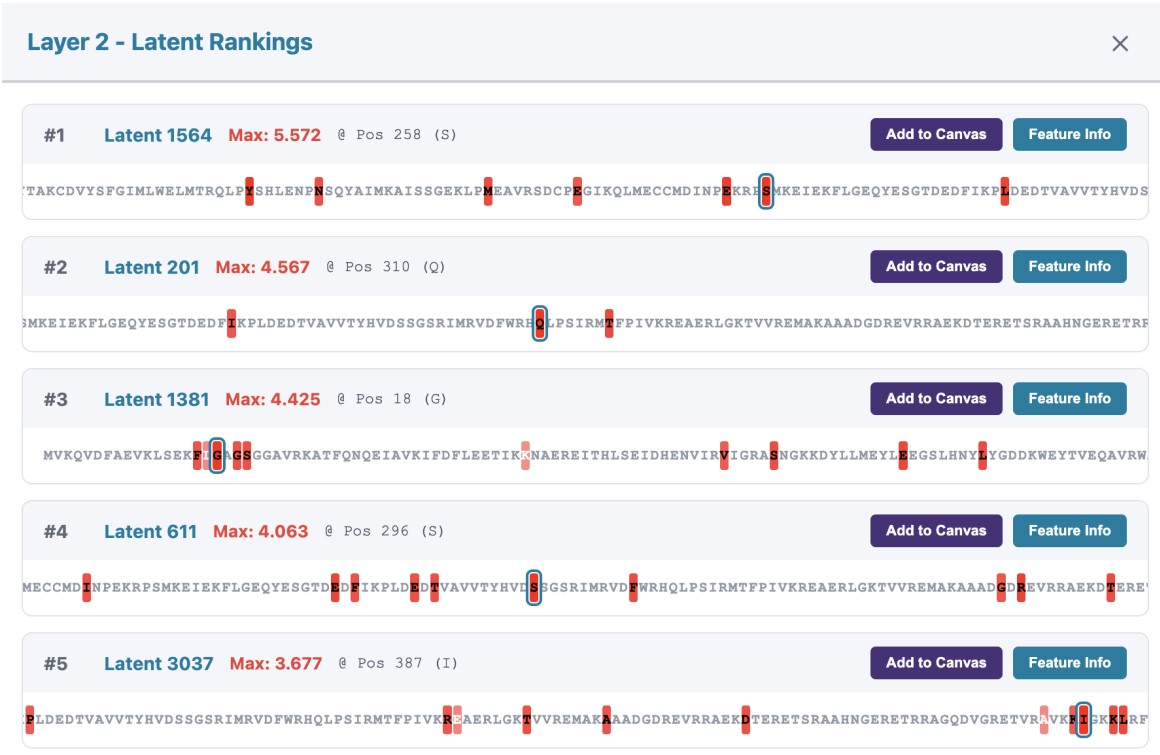

*Figure 10.* **Layer-wise latent rankings.** Clicking on a layer label opens this panel, showing the top-activating latents for that layer ranked by maximum activation value. Each entry displays the latent index, maximum activation score, the sequence position of peak activation, and a sequence window centered on the activation site (highlighted in red). The "Add to Canvas" button enables quick circuit construction, while "Feature Info" opens the detailed latent information panel.

*Table 10.* ESM2 and CLT parameter counts and training times on 1 GPU.

| ESM2 parameters | CLT parameters/Training time (1 GPU) | Windowed CLT parameters/Training time (1 GPU) |
|---|---|---|
| **8M** | 28M/21h | 25M/19h |
| **35M** | 207M/7d | 125M/4d |
| **150M** | 2B/64d | 600M/19d |
| **650M** | 10B/308d | 3B/82d |
| **3B** | 46B/1500d | 11B/360d |
| **15B** | 320B/10000d | 61B/1800d |

To validate this architecture as a scalable alternative, we have trained a windowed CLT on ESM2-35M, which reduced our parameter count by 40% and resulted in a $1.75\times$ speedup in training. On family circuit discovery, we are able to recover 82% of the performance, surpassing the PLT (68%) while approximating the performance of the vanilla CLT (85%) (Figure 16). This strikes a fair tradeoff between capturing cross-layer correlation and compute time.

## H. Comparison to Sparse Autoencoders

SAEs differ from ProtoMech in that they are optimized for reconstruction, rather than transcoding computation. Fig. 17 shows a simplistic example of the differences between SAEs and transcoders. An SAE operating at layer $\ell$ is trained to take as an input $\mathbf{y}^\ell$ and reconstruct $\hat{\mathbf{y}}^\ell$. In contrast, ProtoMech is designed to take as an input $\mathbf{x}^\ell$ and compute $\hat{\mathbf{y}}^\ell$. This means that SAEs, by design, are ignoring the MLP computations inside the transformer, making SAEs insufficient to be replacement model.

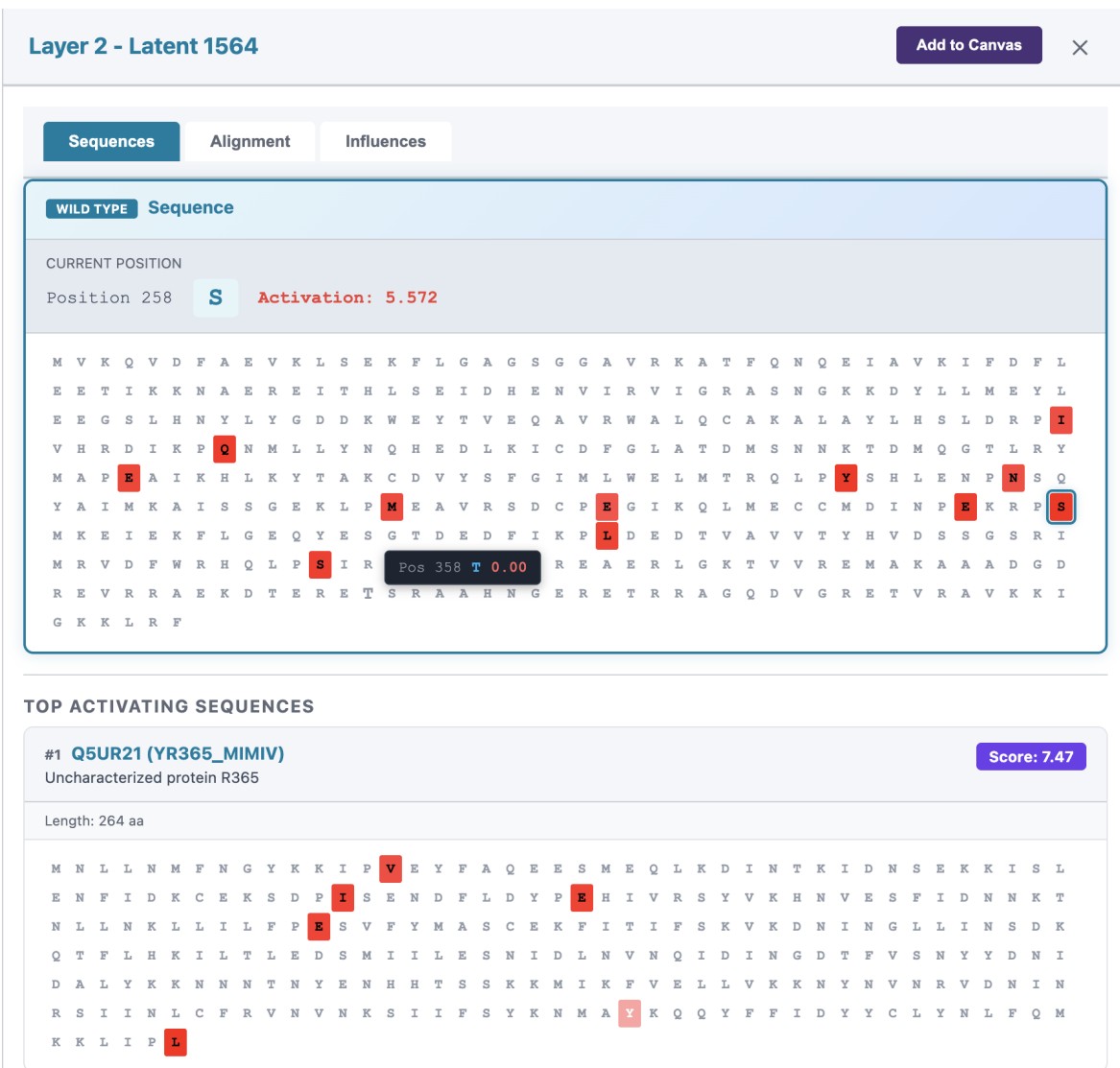

*Figure 11.* **Latent Information Panel: Sequences tab.** The Sequences tab provides detailed activation information for a selected latent. The top section shows the input sequence with activation intensities displayed as a heatmap, where darker colors indicate stronger activation. The current position, amino acid, and activation value are displayed above the sequence. The bottom section lists top-activating sequences from Swiss-Prot, each with its UniProt identifier, protein name, and activation score, enabling users to identify sequence patterns that maximally activate the latent.

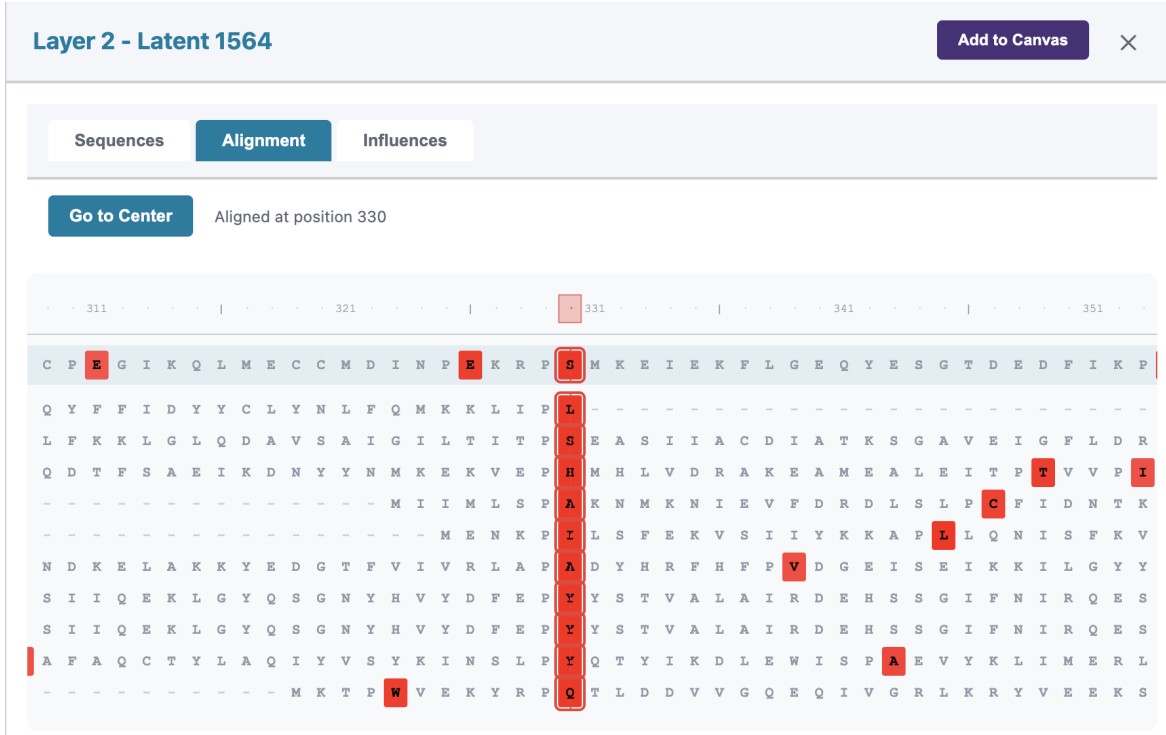

*Figure 12.* **Latent Information Panel: Alignment tab.** The Alignment tab displays an alignment between the input sequence (top row) and top-activating sequences from Swiss-Prot. Activation intensities are shown as colored overlays, with red indicating high activation. The "Go to Center" button navigates to the position of maximum activation of wild type and top activating sequences. This view facilitates identification of conserved sequence motifs and structural features recognized by the latent.

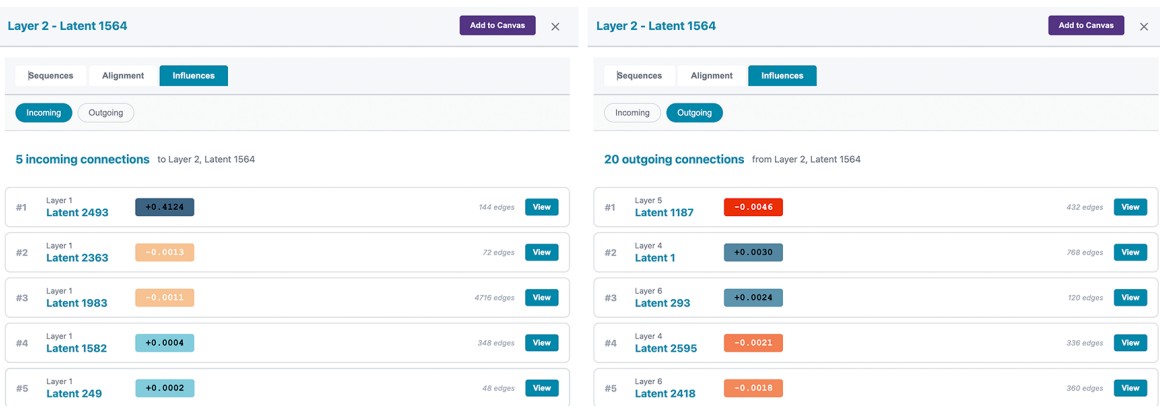

*Figure 13.* **Latent Information Panel: Influences Tab.** The Influences tab reveals the circuit connectivity of a selected latent. The left panel shows incoming connections from earlier layers, while the right panel shows outgoing connections to later layers. Each connection displays the source/target layer and latent index, the virtual weight (green for positive, orange for negative), and the number of sequence positions where this edge is active.

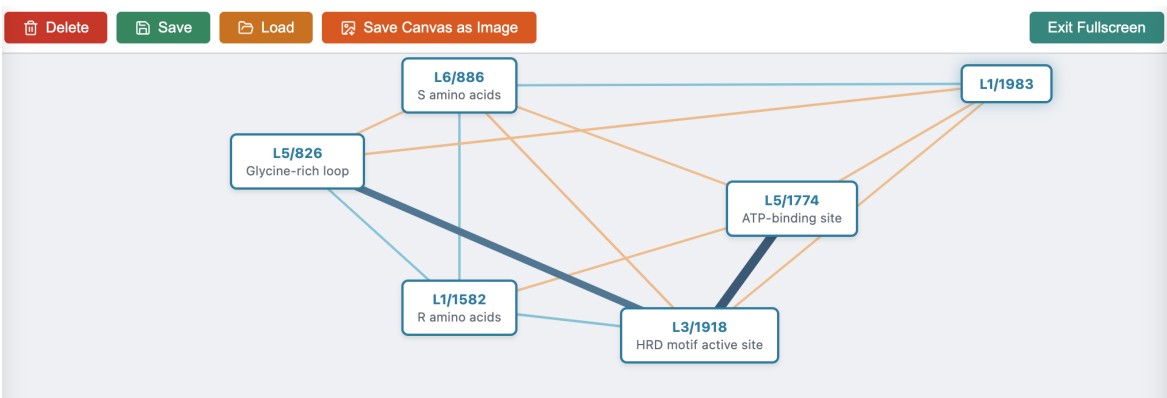

*Figure 14.* **Interactive Circuit Canvas.** The canvas provides a workspace for constructing and annotating circuit diagrams. Nodes represent latents (labeled as Layer/Index, e.g., "L5/1774") and can be annotated with descriptive names documenting their hypothesized function (e.g., "ATP-binding site", "Glycine-rich loop"). Adding a name to a latent can be done by right-clicking the latent. Edges represent virtual weights between latents, with blue indicating positive weights and orange indicating negative weights; edge thickness is proportional to weight magnitude. The toolbar provides functions for deleting nodes, saving/loading canvas layouts, exporting images, and toggling fullscreen mode.

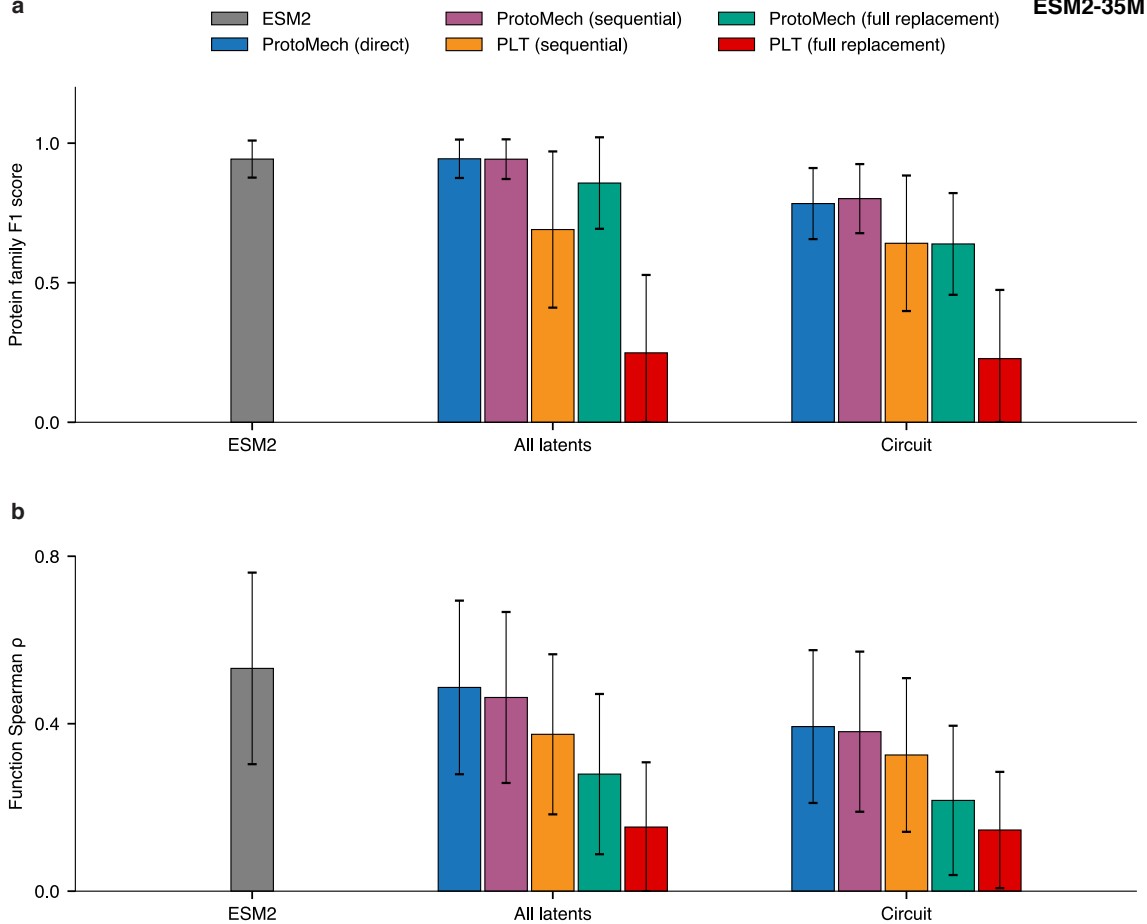

*Figure 15.* Performance of all ESM2-35M replacement models on **a**, protein family classification and **b**, function prediction tasks.

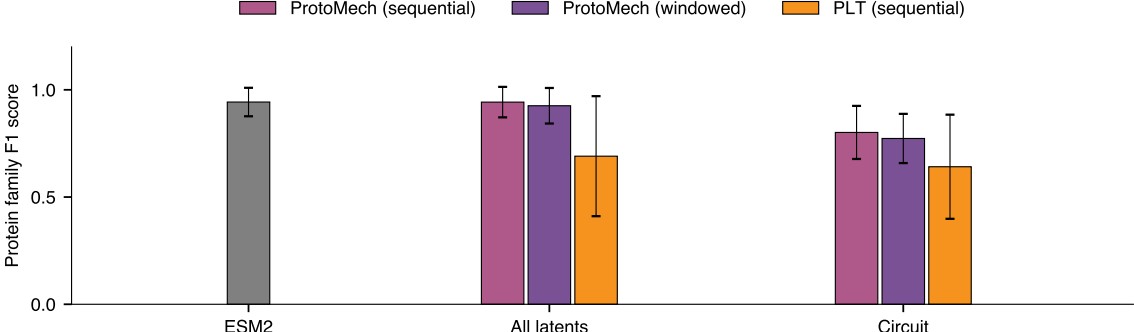

*Figure 16.* Performance of sequential, windowed, and PLT sequential ESM2-35M replacement models on protein family classification tasks.

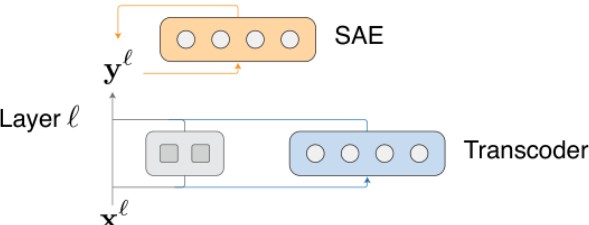

*Figure 17.* A simplified schematic of the differences between SAEs and transcoders. SAEs are designed to take in $\mathbf{y}^{\ell}$ and reconstruct $\hat{\mathbf{y}}^{\ell}$. Transcoders are designed to take in $\mathbf{x}^{\ell}$ and compute $\hat{\mathbf{y}}^{\ell}$.

