# OpenReview forum: "Protein Circuit Tracing via Cross-layer Transcoders"
_ICML.cc/2026/Conference — ICML 2026 regular_

### Official Review · Reviewer_PSZb · 2026-03-10

**Soundness:** 3
**Presentation:** 3
**Significance:** 2
**Originality:** 2
**Overall Recommendation:** 4
**Confidence:** 3

**Summary:**

This paper introduces ProteoMech, a framework designed to uncover the internal computational pathways within protein language models.   While pLMs are highly effective at predicting protein structures and functions, their internal decision-making processes have largely remained opaque. Existing interpretability methods like SAE attempt to decompose these models but treat each layer in isolation, failing to capture how computation accumulates and transforms across multiple layers. ProtoMech employs cross-layer transcoders. It calculate a layer's output as a function of the sparse latent variables from all preceding layers.

**Compliance With Llm Reviewing Policy:**

Affirmed.

**Final Justification:**

My concerns have been adequately addressed.

**Key Questions For Authors:**

How do you propose to mitigate this quadratic parameter scaling bottleneck when applying ProtoMech to deeper, state-of-the-art protein language models?

Have you conducted any preliminary evaluations to determine whether the identified circuits represent global functional rules of the protein family, or if they are merely capturing highly localized epistatic approximations that break down in more distant regions of the fitness landscape?

**Limitations:**

yes

**Strengths And Weaknesses:**

Strengths:

1. The paper is well-organized and clearly written.
2. The proposed method is quite interesting and simple. The use of CLTs to resolve the limitations of independent per-layer approximations is theoretically sound.
3. The authors compare ProtoMech against sensible baselines. Evaluation metrics are standard and appropriate: F1 scores for classification and Spearman rank correlation for fitness prediction.

Weakness:

1. The paper only demonstrates the method on the ESM2-8M model, which contains 6 layers.

---

> ### Author Rebuttal · Authors · 2026-03-29
>
> We thank reviewer PSZb for their comments. Below, we address the following questions:
> > The paper only demonstrates the method on the ESM2-8M model, which contains 6 layers.
>
> As mentioned in our responses to reviewers 19gg and DBJG, to address this concern, we scaled up ProtoMech to ESM2-35M, following a similar methodology outlined in the Appendix. We report our circuit discovery results below (mirroring our analysis from Figs 1c and d):
> **Task (ESM2 performance)**|Method|All latents|Circuit
> -|-|-|-
> |Family (0.94 ± 0.07 F1)|ProtoMech/PLT|**0.94 ± 0.07**/0.69 ± 0.28|**0.80 ± 0.12**/0.64 ± 0.23
> |Function (0.53 ± 0.23 Spearman)|ProtoMech/PLT|**0.46 ± 0.20**/0.38 ± 0.19|**0.38 ± 0.19**/0.33 ± 0.18
>
> Overall, scaling ProtoMech results in increasing ESM2’s performance and the quality of our discovered circuits. The scaled-up version of ProtoMech is available at https://anonacc2026.github.io/. We will add the results above to the camera-ready version.
>
> > How do you propose to mitigate this quadratic parameter scaling bottleneck when applying ProtoMech to deeper, state-of-the-art protein language models?
>
> Thank you for your question. As mentioned in our response to reviewer DBJG, adding cross-layer connections achieves up to a 20\% increase in performance. However, scaling ProtoMech beyond ESM2-35M would likely require a substantial amount of compute. To combat this, we propose “windowed CLTs”, a strategy inspired by [1] where we approximate cross-layer connectivity in localized windows. For example, in ESM2-35M, which has 12 layers, rather than mandating the reconstruction of $\mathbf{y}^{12}$ be a function of layers 1 through 12, we can instead constrain the reconstruction to a localized window of the four immediate preceding layers. We refer to the last Table in reviewer 19gg for estimated training times.
> To validate this architecture, we have trained a windowed CLT on ESM2-35M, which reduces our parameter count by almost 40\% and results in a 1.75$\times$ speedup in training. On family circuit discovery, we are able to recover 82\% of the performance, surpassing the PLT (68\%) while approximating the performance of the vanilla CLT (85\%). This strikes a fair tradeoff between capturing cross-layer correlation and compute time and enables scaling ProtoMech to larger pLMs. We will add a summary of this to the Discussion in the camera-ready.
> > Have you conducted any preliminary evaluations to determine whether the identified circuits represent global functional rules of the protein family, or if they are merely capturing highly localized epistatic approximations that break down in more distant regions of the fitness landscape?
>
> To better address this question, we analyzed the GFP\_AEQVI\_Sarkisyan\_2016 DMS assay, which contains variants up to 10 mutations away. We then stratified the Spearman correlation by mutation depth between ESM2-8M, ProtoMech with all latents, and the circuit discovered:
> |Mutation depth|ESM2|All latents|Circuit
> |-|-|-|-
> 1|0.24|0.16|0.18
> 2|0.30|0.20|0.17
> 3|0.37|0.25|0.23
> 4|0.46|0.34|0.32
> 5+|0.35|0.27|0.26
>
> We repeated this process again on our scaled up ProtoMech on ESM2-35M:
> |Mutation depth|ESM2-35M|All latents|Circuit
> |-|-|-|-
> 1|0.21|0.18|0.15
> 2|0.29|0.24|0.16
> 3|0.37|0.32|0.23
> 4|0.40|0.34|0.25
> 5+|0.31|0.29|0.25
>
> We observe that the circuits discovered by ProtoMech are able to maintain their predictive power across all mutation depths. In particular, at 5+ mutations, ProtoMech circuits recover up to 74\% and 81\% of the predictive power of ESM2-8M and ESM2-35M, respectively, suggesting that ProtoMech is capturing computational mechanisms representative of global functional rules.
>
> We will add these results to the Appendix for the camera-ready version.

---

> > ### Author Rebuttal · Reviewer_PSZb · 2026-04-03
> >
> > My concerns have been adequately addressed.

---

> > > ### Author Response · Authors · 2026-04-03
> > >
> > > We thank the reviewer again for the thoughtful review and for acknowledging that our responses addressed the main concerns.

---

### Official Review · Reviewer_1Xrd · 2026-03-11

**Soundness:** 2
**Presentation:** 3
**Significance:** 2
**Originality:** 2
**Overall Recommendation:** 3
**Confidence:** 4

**Summary:**

The paper proposes ProtoMech, a framework for discovering computational circuits in protein language models using cross-layer transcoders that learn sparse latent representations across layers to capture the model’s computational circuitry. ProtoMech takes EMS2 as the original model, and did experiments on protein family classification task, circuit prediction task and protein sequence optimization task.

**Compliance With Llm Reviewing Policy:**

Affirmed.

**Key Questions For Authors:**

See above

**Limitations:**

See above

**Strengths And Weaknesses:**

# Strengthens
The paper is easy to follow.

# Weaknesses
## Practical utility and experimental issues:
The practical utility of this paper is kind of limited:

1. On all protein family classification task, circuit discovery task and circuit function prediction task, the proposed ProtoMech performs worse than the original model. If the proposed method cannot even beat the original model, even though we might have better interpretation about the model behavior, it isn't seem useful.

2. How are the circuit discovery and function prediction circuits tasks realized? The paper didn't have a clear clarification for this part.

3. For protein mutation towards high fitness task, the paper should also compare to the current SOTA model like PEX [1], GGS[2]. Otherwise, we are unsure about how good the ProtoMech is. Also, how did the fitness score calculated for each DMS dataset, that is to say, what oracle model did the author use to predict the fitness scores?

[1] Proximal Exploration for Model-guided Protein Sequence Design.

[2] Improving Protein Optimization with Smoothed Fitness Landscapes.

4. Also for revealing the known motifs, the sequence is already known and the motifs are also known. The author can interpret the motifs according to the model's behavior. However, if we are given a de novo designed protein sequence, can we still use ProtoMech to uncover the motifs and how can we verify that?

---

> ### Author Rebuttal · Authors · 2026-03-29
>
> We thank reviewer 1Xrd for taking the time to review our paper. We address the following questions:
>
> > If the proposed method cannot even beat the original model, even though we might have better interpretation about the model behavior, it isn't seem useful.
>
> We clarify that the goal in mechanistic interpretability is to find a replacement model that is more human explainable, and often this benefit comes with a natural cost of dropping in accuracy. For example, work by Anthropic applying CLTs to an 18 layer language model [1] shows that CLTs recover 50\% of the original performance. This is not surprising, as the replacement model serves as an interpretable proxy for the original model. Regardless, in the Discussion section, we detail two applications where ProtoMech provides information even beyond the original model. First, ProtoMech *outperforms* the performance of ESM2 when the performance of ESM2 is poor. This suggests that ProtoMech is denoising the performance of ESM2 and distilling the computation down to ESM2’s most salient features. This may have relevance toward regularizing the performance of ESM2. Second, the ability of ProtoMech to provide interpretable explanations of high- and low-fitness mutations (see Fig. 4) may have relevance in high-throughput protein screening. In such applications, ProtoMech can act as a mechanistic filter to select for candidates with plausible biological motifs and prioritize them for wet-lab synthesis.
>
> > How are the circuit discovery and function prediction circuits tasks realized?
>
> Briefly, circuit discovery involves training a supervised probe on the last MLP output, $\mathbf{y}^L$. This serves to establish the performance of ESM2 on family classification and function prediction tasks. ProtoMech then constructs a replacement model for ESM2, where at each layer, it substitutes the original MLP layer with a sparse, interpretable latent representation that approximates the original computation to achieve $\hat{\mathbf{y}}^L$. Circuit discovery aims to isolate the subset of latents in ProtoMech that are governing ESM2’s performance.
>
> We have detailed our full methodology in Appendix D. To address the reviewer's concern, we will add additional explanations in the main text to make this clearer.
>
>
> > For protein mutation towards high fitness task, the paper should also compare to the current SOTA model, like PEX [1], GGS[2]. Otherwise, we are unsure about how good the ProtoMech is.
>
> We clarify that the primary contribution of this paper is focused on understanding the computational mechanisms present in ESM2. Specifically, we compare ProtoMech against per-layer transcoders (PLTs) because, like ProtoMech, it aims to disentangle the model’s computation. We additionally benchmark against Contrastive Activation Addition (CAA) because it also seeks to identify internal representations that govern functional properties. Conversely, since algorithms such as PEX and GGS treat the underlying model as a black-box function, they are not designed to uncover or explain the model's internal computational logic and are out of the scope of this study.
>
> > Also, how did the fitness score calculated for each DMS dataset, that is to say, what oracle model did the author use to predict the fitness scores?
>
> As stated in lines 213-216 of Section 2.3 and Appendix D.3, we trained a CNN evaluation model on 90\% of the training data, following the same training procedure of our supervised models.
>
> > Also, for revealing the known motifs, the sequence is already known, and the motifs are also known. The author can interpret the motifs according to the model's behavior. However, if we are given a de novo designed protein sequence, can we still use ProtoMech to uncover the motif,s and how can we verify that?
>
> We respectively disagree that when "the sequence is already known… the motifs are also known.” Even in natural sequences, the motifs and (more related to our work) the computational circuits ESM2 uses for its predictions are unknown. Furthermore, we clarify that the central claim of the paper is to uncover the computational pathways ESM2 uses to make its predictions. We justify this claim in three sections. In Section 3.1, ProtoMech’s discovered computational pathways that recover the performance of ESM2. In Section 3.2, said pathways can be steered, demonstrating that we discover pathways that ESM2 attributes to protein function. Lastly, in Section 3.3, these pathways overlap with established biological motifs, further corroborating our claim. In the case of de novo protein sequences, ProtoMech’s framework for discovering computational circuits can be applied seamlessly, although uncovering overlapping biological motifs may require more domain expertise.
>
> [1] Ameisen et al. "Circuit Tracing: Revealing Computational Graphs in Language Models" (2025)

---

> > ### Author Rebuttal · Reviewer_1Xrd · 2026-04-04
> >
> > I would still like to see the paper  compare with the current SOTA models on higher fitness tasks. However, the authors didn't provide the results. So I'll keep my current score.

---

> > > ### Author Response · Authors · 2026-04-05
> > >
> > > We appreciate the reviewer’s interest in downstream fitness, but we emphasize that ProtoMech is not designed as a black-box optimization method such as PEX or GGS. Its objective is to recover and manipulate internal computational circuits. As such, comparisons solely framed around optimization performance do not reflect the primary contribution of the work.
> > >
> > > Nevertheless, a meaningful test of the recovered circuits is whether they encode functionally relevant mechanisms. To directly examine this, we conducted an additional analysis on the SPG1 and HIS7 DMS assays. We ran PEX and GGS for 10 rounds under default settings using the respective circuit discovery CNN models as the black-box fitness function (see Appendix D.2). We then evaluated the fitness of the top 50 sequences from each method using our evaluation CNN model following the protocol in Appendix D.3. We report our results in the Table below. The outcome is clear: sequences generated via ProtoMech-based circuit steering outperform those produced by both PEX and GGS across all reported metrics. This is notable given that ProtoMech is not designed as an optimizer, yet it yields superior functional sequences. These results demonstrate that the circuits identified by ProtoMech are not only interpretable but also capture actionable, functionally meaningful mechanisms.
> > >
> > > We will include these results in the camera-ready.
> > >
> > > | Method | DMS | Mean score | Max score | Top 10% score | Top 20% score |
> > > |---|---|---:|---:|---:|---:|
> > > | ProtoMech | SPG1 | **1.67 ± 0.67** | **3.24** | **3.00 ± 0.21** | **2.70 ± 0.34**
> > > | ProtoMech | HIS7 | **1.28 ± 0.09** | **1.42** | **1.41 ± 0.01** | **1.40 ± 0.02**
> > > | PEX | SPG1 | 1.50 ± 0.32 | 2.37 | 2.12 ± 0.16 | 1.95 ± 0.22
> > > | PEX | HIS7 | 0.86 ± 0.10 | 1.09 | 1.04 ± 0.03 | 1.01 ± 0.04
> > > | GGS | SPG1 | 0.65 ± 0.40 | 1.33 | 1.25 ± 0.05 | 1.19 ± 0.08
> > > | GGS | HIS7 | 0.74 ± 0.06 | 0.87 | 0.84 ± 0.02 | 0.82 ± 0.03

---

### Official Review · Reviewer_DBJG · 2026-03-12

**Soundness:** 3
**Presentation:** 3
**Significance:** 2
**Originality:** 2
**Overall Recommendation:** 4
**Confidence:** 4

**Summary:**

This paper introduces ProtoMech, a mechanistic interpretability framework for protein language models (pLMs) that aims to recover computational circuits rather than only layerwise features. The idea of crosscoder is used in LLM interpretability, and this is the first study to use transcoder/crosscoder to interpret the features of the pLMs. ProtoMech is evaluated on (1) protein family classification with logistic probes and (2) fitness/function prediction using ProteinGym DMS assays with CNN probes. The authors also do steering on sequence generation, and have a better performance than CAA.

The finding "discovered circuits occasionally outperform the performance of the full ESM2 model on both protein family classification and function prediction" is interesting.

**Compliance With Llm Reviewing Policy:**

Affirmed.

**Final Justification:**

The author has addressed most of my concerns, and i have increased the score from 3 to 4.

**Key Questions For Authors:**

Some of my concerns/questions

1. Generalization to bigger ESM2 variants or other pLMs： see Weakness 2
2. Cross-layer vs within-layer contribution: Empirically, how much of the gain comes from cross-layer decoding vs simply more parameters / better training?
3. Circuit stability across inputs: If you discover a circuit for a family/assay, how stable is that circuit across different sequences within the family (or across folds in ProteinGym)?

**Limitations:**

You may improve the paper by considering the above weakness and questions.

**Strengths And Weaknesses:**

Strongness:
1. Strong empirical story across multiple tasks: Both classification and regression/fitness prediction show consistent gains over the PLT baseline (trained under similar sparsity).
2. Interesting finding: see discussion "Denoising ESM2.", they find that using crosscoder as a proxy, sometimes can gain better performance than full model(ESM2)
3. The experiments on both tasks show substantial performance recovery using a very small fraction of latent units .

Weakness:
1. Scalability of CLT parameters (O($L^2$) decoders): cross-layer connections require decoder matrices for many layer pairs. This is feasible for ESM2-8M (L=6), but may become expensive for larger L / larger models.
2. Lack of evidence of multiple model families: the experiments are only conducted on ESM2-8M, a very small and limited model; more models /families(even autoregressive models and diffusion LM) should be validated.
3. Faithfulness is partial and somewhat task-probe dependent: recovery is measured relative to probes on the final MLP output (logistic/CNN). This is reasonable, but it may not fully capture whether the CLT reproduces the model’s internal causal mechanisms beyond probe performance.

---

> ### Author Rebuttal · Authors · 2026-03-29
>
> We thank reviewer DBJG for their comments. Below, we address the following questions:
>
> > Generalization to bigger ESM2 variants or other pLMs
>
> We agree that generalization to larger ESM2 variants is an important and open challenge toward the broad application of mechanistic interpretability. To address this question, we have expanded our evaluation.
>
> First, we scaled up ProtoMech to ESM2-35M, following a similar methodology outlined in the Appendix. We report our circuit discovery results below, mirroring our analysis from Figs. 1c and d. ProtoMech successfully identifies high-quality circuits, with performance mirroring the trends observed in the 8M model.
>
> **Task (ESM2 performance)**|Method|All latents|Circuit
> -|-|-|-
> |Family (0.94 ± 0.07 F1)|ProtoMech/PLT|**0.94 ± 0.07**/0.69 ± 0.28|**0.80 ± 0.12**/0.64 ± 0.23
> |Function (0.53 ± 0.23 Spearman)|ProtoMech/PLT|**0.46 ± 0.20**/0.38 ± 0.19|**0.38 ± 0.19**/0.33 ± 0.18
>
> We additionally observe a similar denoising effect mentioned in the Discussion. In family circuits where the performance of ESM2 is low (0.49 ± 0.03), ProtoMech circuits achieve an F1 of 0.68 ± 0.23. These results further reinforce the significance of ProtoMech and its ability to offer utility beyond just interpretability. The scaled-up version of ProtoMech is available at https://anonacc2026.github.io/.
>
> Second, while the vanilla CLT scales poorly with model size, we propose “windowed CLTs”, which approximate cross-layer connectivity in localized windows to enable scaling to larger pLMs. For example, in ESM2-35M, which has 12 layers, we constrain the reconstruction of $\mathbf{y}^{12}$ to a localized window of the four immediate preceding layers, rather than using every single layer. We refer to the last Table in Reviewer 19gg for estimated training times.
>
> We trained a windowed CLT on ESM2-35M, which reduced our parameter count by 40\% and resulted in a 1.75$\times$ speedup in training. On family circuit discovery, we are able to recover 82\% of the performance, surpassing the PLT (68\%) while approximating the performance of the vanilla CLT (85\%). This validates windowed CLTs as a fair tradeoff between capturing cross-layer correlation and compute time.
>
> These results validate that ProtoMech works on a reasonably sized model and has the potential to scale to even larger architectures. We will add the results above to the camera-ready version.
>
> > …(even autoregressive models and diffusion LM) should be validated
>
> We agree that adapting ProtoMech to other pLM architectures is an important challenge. However, we clarify that the primary contribution of this paper is establishing the first framework for circuit discovery in pLMs using cross-layer transcoders, and it is best benchmarked on the masked language modelling architecture. As ESM2 is arguably the most standard model for protein function prediction and fitness modeling, it is imperative that any mechanistic interpretability tool first demonstrates success here.
>
> While extending ProtoMech to other architectures, such as autoregressive models, is possible, its application is not trivial. Autoregressive models often require alignment or fine-tuning strategies for effective generation, which fundamentally differ from the supervised probing we utilize for fitness extrapolation. Furthermore, to the best of our knowledge, CLTs have not yet been applied to diffusion models in either NLP or computational biology literature. Therefore, we consider both these extensions to be beyond the scope of our current study. We will add a summary of this to the Discussion.
>
> > Empirically, how much of the gain comes from cross-layer decoding vs simply more parameters / better training?
>
> In ESM2-8M, we observe that, on average, adding cross-layer connections results in a 20\% and 6\% increase in performance in family and function circuits, respectively. In ESM2-35M, adding cross-layer connections results in a 17\% and 10\% increase in performance in family and function circuits, respectively.
>
> > Circuit stability across inputs: If you discover a circuit for a family/assay, how stable is that circuit across different sequences within the family (or across folds in ProteinGym)?
>
> We observe that both family and function circuits are stable across different inputs/folds. In family circuits, randomly sampling five family sequences from the five largest families, we have an 81\% ± 21\% and 79\% ± 15\% overlap in nodes in ESM2-8M and ESM2-35M, respectively. Similarly, in function circuits, across all 5 folds and all DMS assays in ProteinGym and using the smallest circuit as reference, we have a 96\% ± 4\% and 97\% ± 4\% overlap in nodes in ESM2-8M and ESM2-35M, respectively.
>
> We will add these results to the Appendix in the camera-ready.

---

> > ### Author Rebuttal · Reviewer_DBJG · 2026-04-01
> >
> > I will increase the score to 4

---

> > > ### Author Response · Authors · 2026-04-03
> > >
> > > We thank the reviewer again for the thoughtful review and for acknowledging that our responses addressed the questions raised and clarified the main concerns.

---

### Official Review · Reviewer_19gg · 2026-03-13

**Soundness:** 3
**Presentation:** 4
**Significance:** 3
**Originality:** 2
**Overall Recommendation:** 5
**Confidence:** 4

**Summary:**

This paper trains a cross layer transcoder to learn sparse representations over an ESM language model. This has the presumptive advantage of incorporating information from across layers, rather than creating sparse activations within a layer. The authors use the trained transcoder to discover circuits corresponding to biologically relevant phenomena and perform steering towards high fitness sequences.

**Compliance With Llm Reviewing Policy:**

Affirmed.

**Final Justification:**

My main concerns in my initial reading were proper comparison against existing pLMs to check whether the circuit approach was actually useful. I think the authors have done this and provided reasonable interpolations to larger model sizes. The approach is clean, well executed, and DI think will be valuable for the community.

**Key Questions For Authors:**

1. Please confirm: during the replacement model strategy, the post attention, pre MLP activations are imputed in each layer from the "ground truth" activations from a forward pass of the model. Is this accurate?
2. How is the sparse latent dimension selected?
3. Can you clarify the meaning of the word node in the circuit context? Does a node refer to a single vector (i.e., a layer and sequence position) or something else?
4. Can you get some estimates of model sizes / training time for larger ESM models? How much data do you realistically need to identify a circuit?

**Limitations:**

Yes.

**Strengths And Weaknesses:**

This was a well-written and generally well executed paper. I was a priori dubious of this approach -- there is a lot of mechanistic interpretability work being applied to pLMs without good reason -- but I enjoyed reading this paper. The explanations are clear, the method is well motivated, and the findings seem strong. With this flavor of paper, I think it's particularly important to distinguish and separate the motivation + approach from the natural language space, to avoid simply being an implementation of technique from field A towards field B. My comments below reflect this.

**Soundness:** I'm convinced by the experiments comparing CLTs against PLTs. This is a result that we expect, and the comparison here within sparse reconstruction methods seems very reasonable and straightforward. The range of tasks -- protein classification and fitness improvements -- is sufficient in this regime. With regards to sequence design, there should be an additional ablation considering the raw performance of mutations suggested under an ESM language model. In my view, this ablation is critical. ESM language models have previously been shown capable of guiding protein fitness, but "evolutionary fitness" is a vague label. If one application of circuits is the ability to disentangle particular variants of fitness that we care about, this may be particularly valuable. I don't think the results necessarily need to be SOTA (though it seems if the circuits significantly underperform ESM then this would be problematic).  I'd also like to see a couple of other ESM sizes (particularly ESM-650m -- again, I don't expect that one outperforms this model, but as it is the base model most people use it would be good to have a comparison). Note that I think this ablation has already been done -- it's referenced in denoising ESM2 -- so hopefully it isn't any significant additional work.

**Presentation:** Mostly clean.  I would appreciate an additional figure in the appendix with notation matching the main text that shows (1) how the transcoders are trained and (2) how steering / interventions occur. It does not need to be a pretty figure.

**Significance:** I think the real significance of this paper is the potential ability to disentangle model capabilities, which is a real issue in ml for bio. Language models like ESM show different capabilities at different scales (see [1]) in a way that's a little different than existing models. While this is under explored in this paper, it certainly isn't the paper's responsibility to completely solve the problem and I think it represents a very commendable exploration in this direction.

**Originality:** Yes -- this is the first example of cross coders I've really seen and I think they show some interesting results.


[1] Reverse Distillation: Consistently Scaling Protein Language Model Representations

---

> ### Author Rebuttal · Authors · 2026-03-29
>
> We thank reviewer 19gg for their constructive feedback. Below, we address specific questions:
>
> > additional ablation considering the raw performance of mutations suggested under an ESM language model
>
> Thank you for the suggestion. We added this ablation study and observed that ProtoMech consistently outperforms suggested ESM2 mutations, validating our discovered mechanisms. We will add the full Table to the camera-ready.
>
> | Method | DMS | Mean score | Max score
> |---|---|---:|---:|
> | ProtoMech | SPG1 | **1.67 ± 0.67** | **3.24**
> | ProtoMech | HIS7 | **1.28 ± 0.09** | **1.42**
> | ESM2 | SPG1 | -0.72 ± 1.42 | 2.45
> | ESM2 | HIS7 | 0.73 ± 0.24 | 1.28
>
> > I'd also like to see a couple of other ESM sizes
>
> To address this concern, we scaled up ProtoMech to ESM2-35M. We report circuit discovery results below on the sequential replacement model:
> **Task (ESM2 performance)**|Method|All latents|Circuit
> -|-|-|-
> |Family (0.94 ± 0.07 F1)|ProtoMech/PLT|**0.94 ± 0.07**/0.69 ± 0.28|**0.80 ± 0.12**/0.64 ± 0.23
> |Function (0.53 ± 0.23 Spearman)|ProtoMech/PLT|**0.46 ± 0.20**/0.38 ± 0.19|**0.38 ± 0.19**/0.33 ± 0.18
>
> As expected, scaling ProtoMech results in increasing ESM2’s performance and the quality of our discovered circuits. We additionally observe a similar denoising effect mentioned in the Discussion. For family circuits where ESM2’s F1 is <0.55 (0.49 ± 0.03), ProtoMech circuits achieve an F1 of 0.68 ± 0.23. These results further reinforce the significance of ProtoMech and its ability to offer utility beyond just interpretability.
>
> The scaled-up version of ProtoMech is available at https://anonacc2026.github.io/. We will add the results above to the camera-ready version.
>
> > during the replacement model strategy, the post attention, pre MLP activations… [use] “ground truth” activations
>
> For the replacement strategy in the main text (sequential), the post attention, pre MLP activations are *not* ground-truth. The only ground-truth input from the model is the initial pre MLP input $\mathbf{x}^0_\text{pre}$ and the outputs of the attention heads. In all other layers, the post attention, pre MLP activations are computed based on ProtoMech’s approximation of the forward pass ($\hat{\mathbf{x}}^\ell$). For more details, we refer to Appendix C.2.
>
> > How is the sparse latent dimension selected?
>
> We loosely take inspiration from [1] and perform a 10$\times$ expansion factor over the original ESM2 MLP dimension (see Appendix A). In ESM2-8M, the MLP dimension is 320, which means our sparse latent dimension is 3200. With this choice, our validation normalized MSE during training converged to an average of 0.15. We will add a sentence to the Appendix to clarify this.
>
> > Can you clarify the meaning of the word node in the circuit context? Does a node refer to a single vector (i.e., a layer and sequence position) or something else?
>
> A node refers to a latent at a specific layer. We will clarify this in the main text.
>
> > Can you get some estimates of model sizes / training time for larger ESM models?
>
> Following our hyperparameters from Appendix A, we estimate the following model sizes and training times below. We emphasize that these times are estimated on a single NVIDIA RTX A6000 GPU, and parallelization can make this significantly faster. Obviously, the vanilla CLT scales poorly with model size. To combat this, we propose “windowed CLTs”, a strategy inspired by [2] where we approximate cross-layer connectivity in localized windows. For example, in ESM2-35M, which has 12 layers, we constrain the reconstruction of $\mathbf{y}^{12}$ to a localized window of its four preceding layers, rather than using every single layer. The table below shows the training time of both methods.
>
> ESM2 params|CLT params/Training time (1 GPU)|Windowed CLT params/Training time (1 GPU)
> -|-|-
>  |8M|28M/21h|25M/19h
>  |35M|207M/7d|125M/4d
>  |150M|2B/64d|600M/19d
>  |650M|10B/308d|3B/82d
>  |3B|46B/1500d|11B/360d
> |15B|320B|10000d|61B/1800d
>
> To validate this architecture as a scalable alternative, we have trained a windowed CLT on ESM2-35M, which reduced our parameter count by 40\% and resulted in a 1.75$\times$ speedup in training. On family circuit discovery, we are able to recover 82\% of the performance, surpassing the PLT (68\%) while approximating the performance of the vanilla CLT (85\%). This strikes a fair tradeoff between capturing cross-layer correlation and compute time.
>
> > How much data do you realistically need to identify a circuit?
>
> After the supervised probe is trained, it takes at most 128 sequences to discover our circuits (see Appendix D.1). This efficiency stems from our ability to utilize less than 1% of ProtoMech’s latent space to discover circuits, suggesting that we can compress biologically-relevant information.
>
> [1] Simon et al. “InterPLM: discovering interpretable features in protein language models via sparse autoencoders” (2025)
>
> [2] Shu et al. “Bridging the Attention Gap: Complete Replacement Models for Complete Circuit Tracing” (2026)

---

> > ### Author Rebuttal · Reviewer_19gg · 2026-04-02
> >
> > No further questions. Thanks for the thorough response, I've increased my score.

---

> > > ### Author Response · Authors · 2026-04-03
> > >
> > > We're pleased that our response addressed your concerns, and we appreciate your support of our work. Thank you so much for taking the time to review our submission and for your thoughtful review!

---

### Decision · Program_Chairs · 2026-04-30

**Decision:**

Accept (regular)

**Comment:**

This paper introduces ProtoMech, a framework for tracing cross-layer computational circuits in protein language models. The central contribution is clear. Cross-layer transcoders recover substantially more of ESM2's behavior than per-layer baselines, and the resulting circuits are sparse, biologically interpretable, and usable in steering experiments tied to downstream function.

The main limitations are about scope, not basic soundness. The evidence is still centered on small ESM2 variants, and the computational cost of the full cross-layer approach would become substantial at larger scales. There is also a fair question about how much downstream utility should be demanded from an interpretable replacement model when it is not meant to fully match the original model's performance.

These limitations do not outweigh the main contribution. The strongest concern was whether the method still looks convincing once one asks about scale, circuit stability, and stronger downstream comparisons. On that point, the paper is in a better place after discussion. The added results on ESM2-35M, the direct comparisons showing a benefit from cross-layer structure beyond extra capacity, and the stability analyses across inputs and mutation depths make the core claim much more convincing. The added fitness comparisons also make the steering story more convincing. Overall, I lean towards a positive view on this submission with expectation that all the discussions and additions during the rebuttal would be included in the final version.